



**Characterization of the Long-term Radiosonde Temperature Biases in the Lower Stratosphere using COSMIC and Metop-A/GRAS Data from 2006 to 2014**

Shu-peng Ho[1], Liang Peng[1], Holger Vömel[2]

[1] COSMIC Project Office, University Corporation for Atmospheric Research, Boulder, CO, USA

[2] National Center for Atmospheric Research, Boulder, CO, USA

Manuscript for Atmospheric Chemistry and Physics

September 2016

Shu-Peng Ho, COSMIC Project Office, Univ. Corp. for Atmospheric Research, P. O. Box 3000, Boulder CO. 80307-3000, USA (spho@ucar.edu)



## Abstract

Radiosonde observations (RAOBs) have provided the only long-term global *in situ* temperature measurements in the troposphere and lower stratosphere since 1958. In this study, we use consistently reprocessed Global Positioning System (GPS) radio occultation (RO) temperature data derived from COSMIC and Metop-A/GRAS missions from 2006 to 2014 to characterize the inter-seasonal and inter-annual variability of temperature biases in the lower stratosphere for different sensor types. The results show that the RAOB temperature biases for different RAOB sensor types are mainly owing to i) uncorrected solar zenith angle dependent errors, and ii) change of radiation correction. The mean daytime temperature difference ($\Delta T$) for Vaisala RS92 is equal to 0.18 K in Australia, 0.20 K in Germany, 0.10 K in Canada, 0.13 K in England, and 0.33 K in Brazil. The mean daytime $\Delta T$ is equal to -0.06 K for Sippican, 0.71 K for VIZ-B2, 0.66 K for AVK-MRZ, and 0.18 K for Shanghai. The daytime trend of anomalies for Vaisala RS92 and RO temperature at 50 hPa is equal to 0.00 K/5yrs over United States, -0.02 K/5 yrs over Germany, 0.17 K/5yrs over Australia, 0.23 K/5yrs over Canada, 0.26 K/5yrs over England, and 0.12 K/5yrs over Brazil, respectively. Although there still exist uncertainties for Vaisala RS92 temperature measurements over different geographical locations, the global trend of temperature anomaly between Vaisala RS92 and RO from June 2006 to April 2014 is within +/-0.09 K/5yrs globally. Comparing with Vaisala RS80, Vaisala RS90 and sondes from other manufacturers, the Vaisala RS92 seems to provide the best RAOB temperature measurements, which can potentially be used to construct long term temperature CDRs. Results from this study also demonstrate the feasibility to use RO data to correct RAOB temperature biases for different sensor types.





## 1. Introduction

Stable, long-term atmospheric temperature climate data records (CDRs) with accurate uncertainty estimates are critical for understanding the impacts of global warming in both troposphere and stratosphere and their feedback mechanisms (Thorne et al., 2011; Siedel et al., 2011). Radiosonde observations (RAOBs) have provided the only long-term global *in situ* temperature, moisture, and wind measurements in the troposphere and lower stratosphere since 1958. Several groups have been using multiple years of RAOB temperature measurements to construct long term CDRs (e.g., Durre et al., 2005; Free et al., 2004, 2005; Sherwood et al., 2008; Haimberger et al., 2008, 2011; Thorne et al., 2011; Siedel et al., 2009). However, it has also long been recognized that the measurement quality varies for different sensor types and height (e.g. Luers and Eskridge, 1995, Luers 1997, Luers and Eskridge 1998). Therefore, beside some sensor types where a relatively objective radiation correction had been applied to some sensor types (i.e., Vaisala RS90), it is very difficult to objectively identify, trace, and remove most of the sensor type dependent biases for the historical sonde data and use the corrected RAOB temperatures to construct consistent temperature CDRs. The large uncertainties among temperature CDRs constructed from satellite and in situ measurements are still one of the most challenging issues for global changing researches (IPCC AR5).

The causes of temperature errors among RAOB sensor types include the changing of instruments and practices (Gaffen, 1994) and measurement errors occurring due to the influence of solar and infrared radiation on the thermistor. In the past decade, many homogenization methods were proposed to identify and correct sonde errors due to changing of instruments and practice (Luers and Eskridge 1998; Lanzante et al., 2003;



Andrae et al., 2004; Free et al., 2004, 2005; Sherwood et al., 2008; Haimberger et al.,
2008, 2011; Thorne et al., 2011; Siedel et al., 2009). Possible temperature errors due to
changing of instruments were identified by comparing with those temperature
measurements from adjacent weather stations. However, this approach is limited by the
low number of co-located observations and large atmospheric variability. In addition, due
to lack of absolute references, the remaining radiation temperature biases from adjacent
stations may not be completely removed. As a result, only relative temperature
differences of a possibly large uncertainty among stations are identified.

To correct possible RAOB temperature errors due to radiative effects, Andrae et

al., (2004) and Haimberger et al., (2007, 2008, 2011) used reanalyses data to identify
temperature anomalies between observations and backgrounds, which are then used to
minimize the differences between daytime and nighttime temperature anomalies.
Nevertheless, because changes of reanalysis systems and possible incomplete calibration
of satellite instruments may complicate the temperature anomaly correction, long-term
stability of the derived temperature trends is still of great uncertainty. To correct the
RAOB solar/infrared radiation errors, radiation correction tables (for example, RSN96,
RSN2005 and RSN2010 tables from Vaisala) were introduced by manufactures to correct
for radiation errors of particular sensors. However, when and how exactly different
countries start to apply these corrections and whether there are remaining uncorrected
radiative effects over different geographic regions are still unknown. It is critically
important to use stable and accurate temperature references to characterize these errors
from multiple sensors in different geographical regions over a long period of time.

Unlike RAOBs, the fundamental observable (i.e., time delay) for the Global





Positioning System (GPS) radio occultation (RO) satellite remote sensing technique can
be traced to ultra-stable international standards (atomic clocks) on the ground. The RO
derived atmospheric variables have been used as reference to identify RAOB sensor
dependent biases. For example, Kuo et al., (2004) used RO data to identify sensor type
dependent refractivity biases. Ho et al., (2010a) demonstrated that RO derived water
vapor profiles can be used to distinguish systematic biases among humidity sensors. He et
al., (2009), hereafter He2009 and Sun et al. (2010, 2013) used GPS RO temperature data
in the lower stratosphere to quantify the temperature biases for several sensor types.
While He2009 used the FORMOSAT-3/Constellation Observing System for Meteorology,
Ionosphere, and Climate (COSMIC) post-processed temperature profiles from August
2006 to February 2007 to quantify the radiosonde radiation temperature biases for
different sensor types, Sun et al., (2010; 2013) used COSMIC real-time processed
temperature profiles from April 2008 to August 2009 and from May 2008 to August 2011,
respectively, to identify radiosonde temperature biases for numerical weather prediction
analysis. Because the complete GPS orbital information is not available in a real-time
mode, approximate GPS orbital information was used in the real-time inversion
processing. The differences between real-time and post-processed RO temperatures in the
lower stratosphere range from 0.3 K to 0.1 K depending on the comparison periods (not
shown). Although the real-time COSMIC data, which are processed by using periodically
revised inversion packages, may be suitable for weather analysis, it may not be suitable
for climate studies. Both of these RAOB-RO comparisons are constructed from a
relatively limited period of time. A consistent validation of the variability of inter-
seasonal and inter-annual RAOB temperature biases in a longer time period (close to ten



years) for different temperature sensor types has not yet been done.
Recently, the UCAR Constellation Observing System for Meteorology,
Ionosphere, and Climate (COSMIC) Data Analysis and Archive Center (CDAAC) has
developed an improved reprocessing package, which is used to consistently process RO
data from multiple years of multiple RO missions including COSMIC (launched in April
2006) and Meteorological Operational Polar Satellite–A (Metop-A)/GRAS (GNSS
Receiver for Atmospheric Sounding (launched in October 2006). A sequence of
processing steps is used to invert excess phase measurement to retrieve atmospheric
variables including bending angle, refractivity, pressure, temperature, and geo-potential
height. Brief description of the improved reprocessing package and the general quality of
GPS RO data for climate as benchmark references for climate studies are described in
Appendix A.
The objectives of this study are to use consistently reprocessed GPS RO
temperature data to characterize i) solar zenith angle (SZA) dependent temperature biases,
ii) potential residual temperature errors due to incomplete radiation correction, iii)
temperature biases due to change of radiation correction over different geographical
regions, iv) the inter-seasonal and inter-annual variability of these temperature biases,
and v) the trend analysis and their uncertainty for different senor types in the lower
stratosphere. Contrasting to previous studies (i.e., He2009 and Sun et al. 2010, 2013),
close to 8 years (from June 2006 to April 2014) of consistently reprocessed temperature
profiles derived from COSMIC (i.e., COSMIC2013 covering from June 2006 to April
2014) and Metop-A/GRAS (i.e., Metop-A2016 covering from to September 2007 to
December 2015) are used. Because COSMIC contains dominate sample numbers (six



receivers) than those of Metop-A (one receiver), we limited this study from June 2006 to
April 2014 (see Section 2.2). Because the quality of RO data in the lower stratosphere
does not change during the day or night and is not affected by clouds (Anthes et al.,
2008), the RO temperature profiles co-located with RAOBs are useful to identify the
variation of temperature biases of different temperature sensors. The trend uncertainties
for specific RAOB senor types are also specified. Systematic RAOB temperature biases
and their uncertainties relative to RO data are documented for different sensor types,
which in turn is useful to quantify uncertainties of temperature CDRs constructed from
different RAOB sensor types.
In Section 2, we describe the RO and RAOB data and the comparison method.
The global comparison of RO-RAOB pairs for different temperature sensor types for
daytime and nighttime are summarized in Section 3. The global SZA dependent
temperature biases for various sensor types at different geo-graphical regions are also
compared in this section. The inter-seasonal variations of RAOB-RO temperature biases
are assessed in Section 4. We perform the trend analysis for temperature biases among
sensors in Section 5. We conclude our study in Section 6.

**2. Data and Comparison Method**
**2.1 RAOB data**
The radiosonde data (sonPrf) from June 2006 to April 2014 used in this study are
downloaded from CDAAC (http://cosmic.cosmic.ucar.edu/cdaac/index.html). The sonPrf
data include the temperature, pressure and moisture profiles generated from the original
radiosonde data in the NCAR data archive (http://rda.ucar.edu/datasets/ds351.0), which





provides global radiosonde data with the detailed instrument type.

There are more than 1100 radiosonde stations over both lands and islands globally.

Figure 1 depicts the geophysical locations for all RAOB data from June 2006 to April
2014, which are used in this study. These include Vaisala RS80, RS90, RS92, AVK-
MRZ (and other Russian sondes), VIZ-B2, Shippican MARK II A, Shanghai (from
China), and Meisie (Japan). These radiosonde data are transmitted through the Global
Telecommunication System (GTS). Table 1 summarizes the availability for different
instrument types. In total, seventeen different types of radiosonde systems were used
during the comparison time period. The solar absorptivity ($\alpha$) and sensor infrared
emissivity ($\varepsilon$) for the corresponding thermocap and thermistor for different instrument
types are also summarized in Table 1.  Most of the radiosonde data are collected twice
per day.

Because the Vaisala RS80 sensor was never changed and should be the same

across all RS80 models and the software uses the same radiation correction table which
should not show any differences, we do not further separate Vaisala RS80 sensors (i.e.,
ID=37, 52, 61, and 67). For the same reason, all RS92 sensors (ID=79, 80, 81) are
summarized together and all Russian sensors (ID=27, 75, 88, 89, 58) are also summarized
as AVK sonde (see Table 2 and Section 3.1).

**2.2 GPS RO data**

The re-processed COSMIC (Version 2013.3520, available from June 2006 to

April 2014) and Metop-A/GRAS (Version 2016.0120, available from April 2008 to Dec.
2015)    dry    temperature    profiles    downloaded    from    UCAR    CDAAC



(http://cosmic.cosmic.ucar.edu/cdaac/index.html) are used in this study. With six GPS
receivers on board LEO satellites, COSMIC produced about 1000 to 2500 RO profiles
per day since launch. With one GPS receiver, Metop-A/GRAS produced about 600 RO
profiles per day. To maintain similar sample numbers in each month, we use RO data
from June 2006 to April 2014 in this study. The detail inversion procedures of COSMIC
Version 2013.3520 and Metop-A Version 2016.0120 are summarized in http://cdaac-
www.cosmic.ucar.edu/cdaac/doc/documents/Sokolovskiy_newroam.pdf.    The    general
description of CDAAC inversion procedures is detailed in Kuo et al., (2004), and Ho et
al., (2009a, 2012). In a neutral atmosphere, the refractivity (N) is related to pressure ($P$ in
hPa), temperature ($T$ in K) and the water vapor pressure ($e$ in hPa) according to Smith
and Weintraub (1953):

$$N = 77.6\frac{P}{T} + 3.73*10^5\frac{e}{T^2} \quad .$$
      (1)


Because in the upper troposphere and stratosphere moisture is negligible, the dry
temperature is nearly equal to the actual temperature (Ware et al., 1996). In this study, we
use RO dry temperature from 200 hPa to 20 hPa to quantify the temperature biases for
different sensor types.

**2.3 Detection of RAOB Temperature Biases Using RO Data over Different**
**Geographical Regions**

The RO atmPrf data from COSMIC and Metop-A/GRAS are first interpolated into

the mandatory pressure level of the radiosondes (i.e., 200 hPa, 150 hPa, 100 hPa, 50 hPa,



and 20 hPa). To account for the possible temporal and spatial mismatches between RO
data and RAOBs, the RO atmPrf data within 2 hours and 300 km with those radiosonde
data are collected for different ROAB instrument types, which are similar to the matching
criteria used in He2009. Different from He2009, positions of RO measurements at the
corresponding heights are used in the RAOB-RO ensembles. We compute temperature
differences between RO atmPrf and the corresponding RAOB pairs in the same pressure
level $i$ using the equation

$$\Delta T(i,j) = (1/n) \times \sum_{s=1}^{s=n} \{T_{RAOB}(i,j,s) - T_{RO}(i,j,s)\},$$
(2)


where $j$ is the index for eighteen instrument type listed in Table 1, and $s$ is the index for
all the matched pairs for each of seventeen instrument types.
In addition, we further compare the monthly mean temperature biases for the
matched pairs at different geo-graphical regions. The equation we used to compute the
monthly mean temperature biases between RAOB and RO data at the mandatory height is

$$\Delta T^{Time}(l,m,k) = T_{RAOB}(l,m,k) - T_{RO}(l,m,k),$$
(3)


where $l$, $m$, and $k$ are the indices of the month bin for each vertical grid ($l$), zone ($m$) and
month for the whole time series ($k$ = 1 to 95) from June 2006 to December 2015,
respectively. The geographical zones ($m$) are from USA ($m$=1), Australia ($m$=2),
Germany ($m$=3), Canada ($m$=4), England ($m$=5), Brazil ($m$=6), Russia ($m$=7), and China
($m$=8), Japan ($m$=9), respectively. The standard deviation of the time series ($Std(\Delta T^{Time})$)



is also computed to indicate the variation $\Delta T^{Time}$. In this study, we define daytime data
are from SZA from 0 degree to 90 degree and nighttime data area from SZA from 90
degree to 180 degree. The SZA is computed from the launch time and location of sonde
station since the time and location of the sonde at different height is not available.

**3. Global Mean RAOB Temperature Biases for all Sensor Types Identified by RO**
**Data**
Introduced since 2003, RS92 (ID=79,80,81) from June 2006 to April 2014 was used
in this study. Over oceans, Vaisala RS92 launched from ships were also used (see Figure
1). Since 1981 to the current, Vaisala RS80 (from 1981 to 2014), RS90 (from 1995 to
2014), and RS92 have widely used for numerical weather prediction (NWP) and
atmospheric studies. While the Vaisala have been corrected for possible radiation error
(see RS92 Data Continuity link under Vaisala website), some radiation corrections were
also made for other sensor types although that may not be clearly indicated in the
Metadata files. We quantify the global mean residual radiation correction biases for all
sensor types in this section.

**3.1 The RAOB Temperature Biases during the Daytime and Nighttime for All Senor**
**Types**
In total, we have more than 600,000 RAOB-RO pairs. Using Eq. (2), we compute
the temperature biases of radiosonde measurements for each individual sensor type. The
mean temperature bias for ensembles of the RAOB-RO pairs from June 2006 to April
2014 for the layer between 200 hPa and 20 hPa for different RAOB sensor types is



summarized in Table 2. The standard deviations of the temperature difference ($Std(\Delta T)$)
for the layer between 200 hPa and 20 hPa for each individual radiosonde types are also
calculated.

In general, the radiosonde temperature biases vary for different sensor types. The

mean $\Delta T$ for RS92 (0.16 K), RS80 (0.10 K), RS90 (0.13 K), Sippican MarkIIA (-0.08 K),
Shangai (0.05 K) and Meisei (0.11 K) are smaller than those for AVK (0.33 K) and VIZ-
B2 (0.22 K) (see Table 2).

The solar radiation effect on different sensors is the dominant error source of

RAOB temperature biases (Luers et al., 1998 and He2009). We assume that all
operational data have a radiation correction already applied. The global temperature
biases relative to the co-located RO temperature at 50 hPa for various radiosonde sensor
types for daytime and nighttime are shown in Figures 2a and b, respectively. Only those
stations containing more than 50 RO-ROAB pairs are plotted. Figure 2a depicts that there
exists obvious different mean $\Delta T$ for different sensor types, which vary with geographical
region. Most of the sensor types contain positive temperature biases ranging from 0.1 to
0.5 K during the daytime. This bias during daytime may be a result of the residual error
of the systematic radiation bias correction. Although we only include those stations
containing more than 50 RO-RAOB pairs, some levels of heterogeneity (i.e., Fig. 2a over
Brazil) may be, in part, due to lower sampling numbers. For example, those stations with
temperature biases larger than 0.5 K in the east Brazil contain only about 60 RO-RAOB
pairs. The causes of the temperature biases heterogeneity over North and South China are
not certain in this point.

The mean $\Delta T$ between 200 hPa and 20 hPa for different sensor types in the





daytime is summarized in Table 2. The daytime mean $\Delta T$ (RS92) is about 0.1 K to 0.3 K
globally, and the $\Delta T$ (AVK) is as large as 0.8 K. The relatively larger $Std(\Delta T)$ for
Shanghai (~1.67 K) may be mainly due to large $\Delta T$ difference in north and south China
under different solar zenith angle especially during the daytime. In the daytime $\Delta T$
(Shanghai) can be as large as 0.2 K to 0.4 K in east and north China and range from -0.2
K to -0.4 K in the south China.

The mean nighttime $\Delta T$ is very different from those in the daytime for the same

sensors. Figure 2b shows that most of the sensor types show a cold bias at night except
for Vaisala in South American, Australia, and Europe. The mean $\Delta T$ at night for the two
sonde types with the largest warm bias at daytime (AVK and VIZ-B2) is equal to -0.06 K
and -0.42 K, respectively. The scatter of $\Delta T$ at night is similar for all sonde types with
$Std(\Delta T)$ between 1.55 K and 1.68 K (see Table 2).

The global mean $\Delta T$ for the Vaisala RS92 during the comparison period is slight

larger than the temperature comparison between COSMIC and Vaisala RS92 in 2007 (Ho
et al., 2010b) (~0.01 K) and in He2009 (~0.04 K from ~200 hPa to 50 hPa). This could be
in part because more RS92-RO pairs from lower solar zenith angle regions (for example,
from the southern Hemisphere and near Tropics, see Section 3.2b) are included after
2007 (see section 5). The Stds of $\Delta T$ for daytime and nighttime combined are 1.52 K for
Vaisala RS92, 1.58 K for AVK, 1.67 K for VIZ-B2, 1.59 K for Sippican, 1.68 K for
Shanghai, and 1.69 K for Meisei.

**3.2 Solar Zenith Angle Dependent Temperature Biases for Vaisala Sondes**
**a. Regional Comparison Results**





More than 50% of RAOB data are from Vaisala sondes, from a number of
different countries. In total, 161,019 RS92 (ID=79, 80, 81) ensemble pairs are distributed
in all latitudinal zones during the daytime. To quantify a possible residual radiation
correction error for Vaisala RS92 measurements in the lower stratosphere, which may
vary with SZA, and over different regions, we further compare the mean temperature
differences from 200 hPa to 20 hPa for daytime and nighttime over different regions in
Figures 3 and 4, respectively. Figure 3a is for United States, and Figs. 3b-f are for the
Australia, Germany, Canada, England, and Brazil, respectively.
Figure 3a depicts that RS92 in different regions demonstrate a similar quality in
terms of standard deviation relative to the mean biases when comparing to RO data.
Because some stations in United States are only interested in the tropospheric profiles
and use smaller balloons, less RO-RS92 samples are available above 70 hPa comparing
those in other countries. The $Std(\Delta T)$ (RS92) from 200 hPa to 20 hPa for different
countries are United States (1.59 K), Australia (1.48 K), Germany (1.48 K), Canada (1.43
K), England (1.5 K), and Brazil (1.44 K).
However, there still exist small but not negligible $\Delta T$ (RS92) between 200 hPa
and 20 hPa in different regions. The mean $\Delta T$ (RS92) in the United States is close to zero
near the 200 hPa then slightly increases with height. The mean $\Delta T$ (RS92) in United
States from 200 hPa to 20 hPa is equal to 0.10 K. The mean $\Delta T$ (RS92) are 0.18 K for
Australia, 0.20 K for Germany, 0.10 K for Canada, 0.13 K for England, and 0.33 K for
Brazil (Figs. 3b-e).
Figure 4 depicts the mean RS92-RO temperature differences from 200 hPa to 20
hPa for nighttime, where Figure 4a is for United States, and Figs. 4b-f are for the





Australia, Germany, Canada, England, and Brazil, respectively. The nighttime RS92 data
over different regions show a similar scatter compared to those at daytime. The *Stds(ΔT)*
(RS92) are 1.61 K for United States, 1.50 K for Australia, 1.50 K for Germany, 1.58 K
for Canada, 1.47 K for England, and 1.50 K for Brazil. In most of the regions, the mean
nighttime temperature biases (RAOB minus RO) are 0.1 K to 0.2 K smaller (colder) than
those from the daytime results, except for those in United States and England. The
nighttime mean temperature bias for USA is 0.14 K whereas the daytime mean bias is
0.10 K. The nighttime mean temperature bias for England 0.14 K whereas the daytime
mean bias is 0.13 K. These residual nighttime warm biases are not seen in the ROAB-RO
ensemble pairs for Sippican MARK, VIZ-B2, AVK, and Shanghai Sondes (see Section
3.3). This 0.1 K – 0.2 K warm bias for RS92 at night could be due to calibration of the
RS92 temperature sensor (see Dirksen et al., 2014). The mean nighttime *ΔT* (RS92) are
0.14 K for United States, 0.19 K for Australia, 0.15 K for Germany, -0.10 K for Canada,
0.14 K for England, and 0.25 K for Brazil, respectively (Figs. 4a-e).

The small but not negligible mean Vaisala RS92 temperature biases in different

regions may indicate a small residual error after applying the radiation correction tables
(i.e., RSN96, RSN2005, and RSN2010 tables) for the respective sonde type. Because the
quality of RO temperature is not affected by sunlight in the lower stratosphere, the small
but obvious geographical dependent biases are most likely due to the residual radiation
correction for RS92 and when and how different countries apply the radiation correction
(see Section 4.1). Because all sondes are launched close to the same UTC time, RS92 in
different regions are launched at different local times, i.e. different SZA. The analyses for
the SZA dependent temperature biases are further discussed in next section.



**b. Solar Zenith Angle Dependent Temperature Biases**


To further quantify a possible SZA dependence of the temperature bias due to
residual radiation errors for Vaisala RS92, we bin the computed $\Delta T$ in 5-degree SZA bins
at each of the ROAB mandatory pressure levels above 200 hPa using all the RAOB-RO
ensembles from June 2006 to April 2014. Figure 5 depicts the Vaisala RS92 temperature
biases at 50 hPa as function of SZA where Figure 5a is for the United States, and Figs.
5b-f are for the Australia, Germany, Canada, England, and Brazil, respectively. Only
those bins contains more than 50 RAOB-RO pairs are included. Low SZA is at noon and
90 degrees SZA is for sunrise or sunset, where the solar elevation angle is close to zero.
Figure 5 shows that the mean $\Delta T$ (RS92) has a slightly larger warm bias for low SZA
(near noon) than that at higher SZA (late afternoon and in the night).

**3.3 Solar Zenith Angle Dependent Temperature Biases for Sippican MARK, VIZ-**


**B2, AVK-MRZ, and Shanghai Sondes**


Unlike Vaisala sondes, which are distributed in almost all latitudinal zones, other
sonde types are distributed mainly in the northern mid-latitudes. Figs. 6a-d depict the
mean temperature differences from 200 hPa to 20 hPa in the daytime for $\Delta T$ (Sippican),
$\Delta T$ (VIZ-B2), $\Delta T$ (AVK), and $\Delta T$ (Shanghai), respectively. The mean $\Delta T$ is -0.06 K for
Sippican, 0.71 K for VIZ-B2, 0.66 K for AVK-MRZ, and 0.18 K for Shanghai,
respectively. These mean biases are similar, but not exactly the same as those from
He2009 (not shown). This may owe, in part, to the sampling differences between He2009
(from 2006 August to 2007 Feb., 7 months) and this study (95 months).
Figs. 7a-d depict the mean temperature differences from 200 hPa to 20 hPa in the



nighttime also for $\Delta T$ (Sippican), $\Delta T$ (VIZ-B2), $\Delta T$ (AVK-MRZ), and $\Delta T$ (Shanghai),
respectively. The mean $\Delta T$ is -0.10 K for Sippican, -0.42 K for VIZ-B2, -0.06 K for AVK,
and -0.07 K for Shanghai. Their $Stds(\Delta T)$ are 1.62 K for Sippican, 1.60 K for VIZ-B2,
1.56 K for AVK, and 1.68 K for Shanghai, respectively.

Similar to Vaisala, we also bin the computed $\Delta T$ in 5-degree SZA bins for each

mandatory pressure levels above 200 hPa using all the RAOB-RO pairs from June 2006
to April 2014 for different sensor types. Only those bins contains more than 50 RAOB-
RO pairs are included. Figures 8a-d depicts $\Delta T$ at 50 hPa varying for SZA ranging from 0
degrees to 180 degrees for Sippican MARK, VIZ-B2, AVK-MRZ, and Shanghai,
respectively.

The SZA for Sippican, VIZ-B2, Russian, and Shanghai sondes is mainly ranging

between 30 degree and 150 degree. The VIZ-B2 has an obvious warm bias during
daytime and a cold bias at night relative to those of RO temperature profiles (Figure 8b).
At 50 hPa, the VIZ-B2 warm bias can be as large as 1.75 K near the noon, and it
decreases to -0.8 K during the night. AVK has a temperature bias of from about 0.7 K to
1.1 K in the daytime where its nighttime biases are close to zero (Figure 8c). The mean
temperature biases for the Shanghai sondes is about 0.16 K and -0.07 K for daytime and
nighttime (Figure 6d), respectively.

**4. Comparison of the Seasonal RAOB Temperature Biases in different Regions**

Since there is some residual radiation error, we characterize the long-term

stability of RAOB temperature measurements for different RAOB sensor types by
quantifying their seasonal temperature biases relative to those of co-located RO data.





### 4.1 Identification of RS92 Temperature Biases due to Change of Radiation Correction


The RAOB-RO monthly mean temperature biases in the lower stratosphere at
different geographical regions are highly dependent on the seasonal variation of the SZA.
The Vaisala RS92 radiosonde was introduced in 2003 and is scheduled to be replaced by
the Vaisala RS41 in 2017. Vaisala included a reinforcement of the RS92 sensor in 2007,
which impacted the radiation error. To account for this sensor update, the radiation
correction tables were updated in 2011 (RSN2010, software version 3.64), which is used
to replace the original radiation correction table. Between 200 hPa and 20 hPa, the
correction in RSN2010 is about 0.1 K stronger than in RSN2005 (see
http://www.vaisala.com/en/products/soundingsystemsandradiosondes/soundingdatacontin
uity/RS92DataContinuity/Pages/revisedsolarradiationcorrectiontableRSN2010.aspx). It is
likely that a country updated the correction table for their entire network. However, when
exactly each country implemented these updated Vaisala radiation correct tables is
unknown.
To identify possible RS92 temperature biases due to changes of the radiation
correction table (i.e., RSN2010), we compare the mean $\Delta T$ from January 2007 to
December 2010 (i.e., $\Delta T$ (RS92$_{200701\text{-}201012}$)) to those from January 2011 to April 2014 (i.e.,
$\Delta T$ (RS92$_{201101\text{-}201404}$) over the United States, Australia, Germany, Canada, England, and
Brazil (Figs. 9a-f). RO temperature is used as references for these two periods. Results
show that there is no obvious $\Delta T$ change between these two periods for the RS92 sondes
over United States and Germany (the mean daytime temperature difference between these
two periods are about -0.05 K and -0.01 K in 50 hPa for United States and Germany,





respectively, see Figs. 9a and c). However, the daytime temperature difference between
$\Delta T$ (RS92$_{200701-201012}$) and $\Delta T$ (RS92$_{201101-201404}$) over Australia, Canada, England, and
Brazil show obvious close to 0.1 K to 0.15 K difference varying at different heights (see
Figures 9b, c, e, and f, respectively). Note that over Australia, the temperature difference
between these two periods at 20 hPa is also as large as -0.2 K, which may also be resulted
in the incomplete radiation correction. The incomplete radiation correction likely leads to
small but not negligible anomaly in the time series. In this case, the trend anomalies in
Australia, Canada, England, and Brazil at 50 hPa is larger than those over the United
States and over Germany (see Section 5.2).
The Deutscher Wetterdienst (DWD), Germany's Meteorological Service,
implemented the updated radiation correction for the Vaisala RS92 not in 2011, but in the
spring of 2015, to avoid inconsistencies with corrections already implemented in their
data assimilation system. This may in part explain the better consistency
of $\Delta T$ (RS92$_{200701-201012}$) and $\Delta T$ (RS92$_{201101-201404}$) over Germany than over other
countries. This also indicates the importance of establishing traceability through careful
documentation and metadata tracking, which is especially crucial for radiosonde data
used in climate studies. The relatively small temperature difference between these two
periods over the United States is most likely a statistical artifact due to the very small
number of coincidences in this period, since the US National Weather Service (NWS) did
not use Vaisala RS92 radiosondes before 2012.

**4.2 Time Series Anomaly for RS92**
SZA-dependent biases may result in seasonally and regionally dependent





temperature biases for different sensor types, which may result in unexpected trend
uncertainty. With a residual RS92 radiation error identified in the Section 3.2, the time
series of the RS92-RO temperature bias behave slightly different for different regions.
Figures 10a-f show daytime and nighttime time series of monthly mean temperature
biases computed using Eq. (3) at 50 hPa for $\Delta T^{Time}$ (RS92) at United States ($\Delta T^{Time}$
(RS92$_{USA}$)), Australia ($\Delta T^{Time}$ (RS92$_{Australia}$)), German ($\Delta T^{Time}$ (RS92$_{German}$)), Canada
($\Delta T^{Time}$ (RS92$_{Canada}$)), England ($\Delta T^{Time}$ (RS92$_{England}$)), and Brazil ($\Delta T^{Time}$ (RS92$_{Brazil}$)),
respectively. The number for the monthly RAOB-RO pairs for daytime is in pink dash
line and that for the nighttime is in green dash line. The vertical lines superimposed on
the monthly mean are the standard error of the mean.
Figures 10a-f indicate that the time series of $\Delta T^{Time}$ (RS92) at all regions are largely
constant in time with a small difference during the daytime and nighttime in each
individual regions. The consistency of RAOB and RO time series data is best represented
by their standard deviation. The $Stds(\Delta T^{Time})$ are 0.4 K for United States, 0.18 K for
Australia, 0.20 K for Germany, 0.35 K for Canada, 0.39 K for England, and 0.22 K for
Brazil, respectively. The relatively larger $Std(\Delta T^{Time})$ for United States and England may
be owing to smaller samples (less than 40 RAOB-RO pairs in most of the months from
2006 to 2014). The relative larger $Std(\Delta T^{Time})$ in Canada is mainly caused by the seasonal
sampling difference. During summer, daytime RAOB-RO pairs are as many as 400 and
drop to less than 10 pairs during winter (Figure 10d). The mean daytime temperature
biases are 0.08 K for United States, 0.22 K for Australia, 0.22 K for Germany, -0.06 K
for Canada, 0.12 K for England, and 0.35 K for Brazil.
The $Std(\Delta T^{Time})$ for RS92 at night are larger than those during daytime, except for





those in Canada, which may be due to a relative smaller RAOB-RO ensemble pairs in the
nighttime over those regions. The $Stds(\Delta T^{Time})$ for RS92 are 0.46 K for United States,
0.30 K for Australia, 0.24 K for Germany, 0.32 K for Canada, 0.42 K for England, and
0.43 K for Brazil in the nighttime. Their mean nighttime temperature biases are 0.19 K
for United States, 0.23 K for Australia, 0.21 K for Germany, -0.01 K for Canada, 0.16 K
for England, and 0.26 K for Brazil. The less than 0.5 K $Std(\Delta T^{Time})$ for RS92 over
daytime and nighttime over these six regions actually demonstrate the long-term stability
of RS92 data.

The variation of mean $\Delta T^{Time}$ at different regions is highly related to the

corresponding variation of SZA. The largest mean $\Delta T^{Time}$ (RS92) is over Brazil (i.e.,
$\Delta T^{Time}$ (RS92$_{Brazil}$) see Figure 10f), where the mean $\Delta T^{Time}$ (RS92$_{Brazil}$) is equal to 0.35 K
and 0.26 K for the daytime and nighttime, respectively.

A seasonal variation of $\Delta T^{Time}$ (RS92) is not apparent expect over Canada.

Although the mean temperature biases are very small (less than +/- 0.06 K) over Canada
(i.e., northern high-altitudes), there still exist some seasonal-dependent temperature bias,
which could be a result of the very few RAOB-RO ensemble pairs for night time in
summer, and for daytime night time in winter (Fig. 10d). Over Canada, daytime SZA is
as high as 50 degree in summer which becomes 88 degree during the winter. Therefore,
the daytime $\Delta T^{Time}$ (RS92) can be as large as 0.3 K during the summer and as low as -0.3
K during the winter.

With less radiative effect on sondes, the magnitude of RAOB-RO temperature

bias at150 hPa is in general smaller than those in 50 hPa (see Table 4). The mean $\Delta T^{Time}$
(RS92) at 150 hPa daytime temperature differences are 0.00 K for United States, 0.03 K





for Australia, 0.07 K for Germany, 0.03 K for Canada, 0.04 K for England, and 0.21 K
for Brazil. The corresponding $\Delta T^{Time}$ (RS92) for nighttime for these countries are 0.09 K
for United States, 0.08 K for Australia, 0.06 K for Germany, 0.12 K for Canada, 0.05 K
for England, and 0.23 K for Brazil. The $Std(\Delta T^{Time})$ for RS92 at these six regions are all
less than 0.37 K during the day and less than 0.52 K during the night (Table 4).

**4.3 Time Series Anomaly for Sippican MARK, VIZ-B2, AVK-MRZ, and Shanghai**
**Sondes**

To demonstrate the inter-seasonal and inter-annual variation of the RAOB-RO

temperature biases, the time series of the monthly mean temperature bias for Sippican
MARK IIA (ID=87), VIZ-B2, AVK, and Shanghai (i.e., $\Delta_{Sippican}^{Time}$, $\Delta_{VIZ-B2}^{Time}$, $\Delta_{AVK}^{Time}$, and
$\Delta_{Shanghai}^{Time}$) in the northern mid-latitudes (from 60°N to 20°N) are shown in Figure 11. The
mean temperature biases in this region for $\Delta_{Sippican}^{Time}$, $\Delta_{VIZ-B2}^{Time}$, $\Delta_{AVK}^{Time}$, and $\Delta_{Shanghai}^{Time}$ for 50 hPa
are summarized in Table 5.

Figure 11 shows the time series of the monthly mean temperature bias at 50 hPa.

During daytime $\Delta_{Sippican}^{Time}$ (-0.12 K), $\Delta_{VIZ-B2}^{Time}$ (0.87 K), $\Delta_{AVK}^{Time}$ (0.80 K), and $\Delta_{Shanghai}^{Time}$ (0.10 K)
are warmer than those in the nighttime. The monthly mean temperature biases at 50 hPa
for nighttime are -0.12 K for $\Delta_{Sippican}^{Time}$, -0.56 K for $\Delta_{VIZ-B2}^{Time}$, -0.03 K for $\Delta_{AVK}^{Time}$, and -0.20 K
for $\Delta_{Shanghai}^{Time}$. While $\Delta_{Sippican}^{Time}$ and $\Delta_{Shanghai}^{Time}$ are largely constant in time, $\Delta_{VIZ-B2}^{Time}$ has obvious
seasonal variations with a negative trend during nighttime and positive trend during
daytime. The number of VIZ-B2 observations drops off after 2012 (see Figure 11b),
which contributes to the larger variation of the $\Delta_{ViZ-B2}^{Time}$ after then.



$\Delta^{Time}_{AVK}$ has an irregular seasonal variation particularly during daytime, with a large
warm bias. A part of this irregular bias may be due to an unidentified change of
instrumentation and large seasonal variations in sample numbers. The standard deviation
of the temperature differences for these four sensors (i.e., $Std(\Delta^{Time}_{Sippican})$, $Std(\Delta^{Time}_{VIZ-B2})$,
$Std(\Delta^{Time}_{AVK})$, and $Std(\Delta^{Time}_{Shanghai})$) at daytime are 0.33K for $\Delta^{Time}_{Sippican}$, 0.37 K for $\Delta^{Time}_{VIZ-B2}$, 0.22
K for $\Delta^{Time}_{AVK}$, and 0.18 K for $\Delta^{Time}_{Shanghai}$. The corresponding nighttime variations are 0.21 K
for $\Delta^{Time}_{Sippican}$, 0.43 K for $\Delta^{Time}_{VIZ-B2}$, 0.21 K for $\Delta^{Time}_{AVK}$, and 0.17 K for $\Delta^{Time}_{Shanghai}$. The mean time
series temperature bias at 50 hPa for these sensor types for the northern mid-latitude is
summarized in Table 5 and the corresponding mean time series temperature differences at
150 hPa is summarized in Table 6.

**5. Trend Analysis and Potential Causes of RAOB Temperature Trend Uncertainty**
**5.1 Comparison Method**
To further quantify inter-annual variation of RAOB temperature biases for
different sensor types, we conduct the trend analysis for the time series of RAOB-RO
temperature anomaly. The anomaly of trend for each of individual sensor types relative to
those of co-located RO temperature are computed and compared. We focus on the trend
analysis for individual sensor types over specific regions similar to previous sections. The
de-seasonalized temperature anomalies are computed by:

$$\Delta T^{Deseason}(l,m,k) = T_{RAOB}(l,m,k) - \overline{T^{Time}}(l,m,k^{'}),  \tag{4}$$



where $l$, $m$, and $k$ are the indices of the month bin for each layer ($l$), zone ($m$) and month
for the whole time series ($k$ = 1 to 95), respectively, and $k^{'}$ is the index of the month bin
of the year ($k^{'}$ =1 to 12). $\overline{T^{Time}}(l,m,k^{'})$ is the mean RO temperature co-located for
different sensor types for each level ($l$), zone ($m$), and averaged over all available years
for a particular month ($k^{'}$). Note that because the period of available measurements for
each of the sensor types is different, the months used to compute $\overline{T^{Time}}(l,m,k^{'})$ may vary
for different sensor types. The mean trend of temperature difference anomalies for each
of the sensor types at 50 hPa and 150 Pa are summarized in Tables 5 and 6, respectively.

**5.2 Trend of Temperature Anomalies for Vaisala Sondes**

The trend uncertainty for RAOB over different regions are mainly due to i)

uncorrected solar zenith angle dependent biases, ii) changing of radiation correction, iii)
and iv) small samples used in the trend analysis. While it is not possible to identify the
bias for each of the individual causes, we can only quantify the combined statistical
biases using RAOB-RO ensembles.

Figure 12 depicts the de-seasonalized temperature anomalies for Vaisala RS92

over United States ($\Delta T_{RS92\_USA}^{Deseason}$), Australia ($\Delta T_{RS92\_Australia}^{Deseason}$), German ($\Delta T_{RS92\_Germany}^{Deseason}$),
Canada ($\Delta T_{RS92\_Canada}^{Deseason}$), England ($\Delta T_{RS92\_England}^{Deseason}$), and Brazil ($\Delta T_{RS92\_Brazil}^{Deseason}$), respectively.
In general, daytime trend differences at 50 hPa in all six regions are within ±0.26 (K/ 5yrs,
see Table 3). While the daytime trend of anomalies for RAOB and RO temperature at 50
hPa for United States and Germany are 0.00 and -0.02 K/5yrs, the trend of anomalies are
equal to 0.18 K/5yrs over Australia, 0.24 K/5yrs over Canada, 0.26 K/5yrs over England,





and 0.12 K/5 yrs over Brazil, respectively. This non-trivia trend anomaly in the later
regions may be owing to the incomplete daytime radiation correction applied in these
regions between $\Delta T$ (RS92$_{200701\text{-}201012}$) and $\Delta T$ (RS92$_{201101\text{-}201404}$) (see Figure 9). The
corresponding nighttime trend differences in these six regions are -0.21 K/ 5yrs for
United States, -0.08 K/ 5yrs for Australia, -0.14 K/ 5yrs for Germany, -0.02 K/ 5yrs for
Canada, -0.16 K/ 5yrs for England, and -0.10 K/ 5yrs for Brazil (see Table 3).
To further examine the temperature trend uncertainty for global Vaisala sensors,
we compare the global trend of anomaly for RS80, RS90, and RS92 at 50 hPa and 150
hPa in Tables 5 and 6, respectively. The global de-seasonalized temperature anomalies
for Vaisala RS92 for daytime and nighttime are equal to 0.07 K/5yrs and -0.09 K/5yrs,
respective (Table 5). The 95% confidence intervals for slopes are shown in the
parentheses. This indicates that although there might be a small residual radiation error
for RS92, the trend anomaly between RS92 and RO from June 2006 to April 2014 is
within +/-0.09 K/5yrs globally. These values are just above the 1 sigma calibration
uncertainty estimated by Dirksen et al., (2014). This means that probably the stability of
the calibration alone could explain most of this very small trend. It is also consistent with
the change in radiation correction.
The trend anomaly between RS80 and RO is about 0.19 K/5yrs during the day
and 0.11 K/5yrs during the night (Table 5). Those between RS90 and RO temperature in
the lower stratosphere are equal to -0.01 K/5yrs and 0.04 K/5yrs for daytime and
nighttime, respectively (Table 5).
To compute the degree of deviation between RAOB temperature and RO
temperature, we also calculate the root mean square (RMS) temperature difference of the





581 derived de-seasonalized anomalies. The global RMS for $\Delta T_{RS92}^{Deseason}$ in the daytime is equal

582 to 0.06 K. This indicates the consistency of RS92 temperature measurements relative to

583 the RO temperature. Both of the global RMS for RS80 and RS90 in daytime are (0.27,

584 0.26) K (Table 5).

585  Because the RAOB temperatures in 150 hPa are less biased compared to those at

586 50 hPa, the de-seasonalized temperature anomalies for Vaisala Sondes at 150 hPa are

587 even smaller than those at 50 hPa. The trend differences for $\Delta T_{RS92}^{Deseason}$ at 150 hPa for

588 RS92 are -0.13 K/5yrs for United States, 0.12 K/5yrs for Australia, -0.02 K/5yrs for

589 Germany, 0.23 K/5yrs for Canada, 0.06 K/5yrs for England, and 0.00 K/5yrs for Brazil

590 during the daytime (see Table 4). The corresponding trend differences during the

591 nighttime are -0.23 K/5yrs for United States, -0.07 K/5yrs for Australia, -0.19 K/5yrs for

592 Germany, -0.21 K/5yrs for Canada, -0.08 K/5yrs for England, and -0.01 K/5yrs for Brazil.

593 The global RMS of RAOB-RO anomalies for RS92 at 150 hPa are 0.04K for daytime and

594 0.07 K for nighttime (Table 6).

596 **5.3 Trends of Temperature Anomalies for Sippican MARK, VIZ-B2, AVK-MRZ,**

597 **and Shanghai Sondes**

598  Figure 13 depicts the de-seasonalized temperature anomalies for Sippican MARK

599 IIA (ID=87), VIZ-B2 (ID=51), AVK-MRZ (ID=27), and Shanghai (ID=32) (i.e.,

600 $\Delta T_{MARK-IIA}^{Deseason}$, $\Delta T_{VIZ-B2}^{Deseason}$, $\Delta T_{AVK}^{Deseason}$, and $\Delta T_{Shanghai}^{Deseason}$) at 50 hPa, respectively. The trends of

601 temperature anomalies for these sensor types are listed in Table 5. The 95% confidence

602 intervals for slopes are shown in the parentheses. The daytime temperature trend

603 anomalies are 0.41 K/5yrs for $\Delta T_{MARK-IIA}^{Deseason}$, 0.47 K/5yrs for $\Delta T_{VIZ-B2}^{Deseason}$, -0.14 K/5yrs for





$\Delta T_{AVK}^{Deseason}$, and 0.18 K/5yrs for $\Delta T_{Shanghai}^{Deseason}$, which are much larger than those of the
Vaisala RS92. The corresponding nighttime trend anomalies are 0.24 K/5yrs ($\Delta T_{MARK-IIA}^{Deseason}$),
-0.35 K/5yrs ($\Delta T_{VIZ-B2}^{Deseason}$), -0.14 K/5yrs ($\Delta T_{AVK}^{Deseason}$), -0.02 K/5yrs ($\Delta T_{Shanghai}^{Deseason}$). Since the
number of AVK - RO pairs decrease significantly after 2012, the trend anomaly for
AVK-RO pairs before and after 2012 vary.
The root mean square (RMS) of the de-seasonalized time series $Std(\Delta T^{Time})$ is
used to indicate the trend uncertainty of the time series. The trend differences and RMS
for all the sonde types at 50 hPa and 150 hPa are summarized in Tables 5 and 6,
respectively.

**6. Conclusions and Future Work**
In this study, we used consistently reprocessed GPS RO temperature data to
characterize radiosonde temperature biases and the inter-seasonal and inter-annual
variability of these biases in the lower stratosphere for different radiosonde types. We
reach the following conclusions.
1. Solar zenith angle dependent biases: The solar radiative effect on different sensors
is the dominant error source of RAOB temperature biases during daytime. With the
consistent precision of RO temperature data between COSMIC and Metop-A, we are able
to identify the mean temperature biases from 200 hPa to 20 hPa layer among older
sensors (i.e., Vaisala RS80 sensors with ID=37, 52, 61, and 67), and new sensors (i.e.,
RS92 sensors with ID=79, 80, 81), and the obvious daytime and nighttime biases for the
same sensor types which is usually distributed in the same countries (i.e., Shanghai
sensor in China, AVK in Russian, VIZ-B2 in in United Stated). Because the quality of





RO temperature is not affected by sunlight in the lower stratosphere, those
daytime/nighttime biases shall mainly originate from uncorrected radiation biases for
each individual sensor types. Most of the sensor types contain positive temperature biases
ranging from 0.1 to 0.5 K during the daytime. Among all the sensors, the Vaisila RS92
has the smallest temperature biases in the lower stratosphere comparing to the co-located
RO temperatures. The daytime mean $\Delta T$ (RS92) is about 0.1 K to 0.3 K globally. The $\Delta T$
(AVK) mainly distributed over Russian is as large as 0.8 K. Most of the sensor types
contain cold bias in the night where the mean $\Delta T$ (AVK) and $\Delta T$ (VIZ-B2) in the night
time are as large as -0.22 K and -0.54 K, respectively.

2. Residual solar zenith angle dependent biases: After applying the solar radiation

correction, most of the RS92 daytime biases are removed. However, a small residual
radiation bias for RS92 remains, which varies with different geographical region or
operating organization. Similar to He2009 and Sun et al., (2010, 2013), there exist a
small SZA dependent biases among different sensor types. The global mean residual
temperature biases for RS92 (i.e., $\Delta T$ (RS92) from SZA 0 to 45 degrees in 20 hPa, 50 hPa,
and 150 hPa are close to 0.3 K, 0.15 K, and 0.05 K, respectively. These biases are less
than the uncertainty described in Dirksen et al., (2014). In the daytime, the mean $\Delta T$
(Sippican), $\Delta T$ (VIZ-B2), $\Delta T$ (AVK-MRZ), and $\Delta T$ (Shanghai) are (-0.06, 0.71, 0.66, 0.18)
K.

3. Changing of radiation correction and RAOB temperature uncertainty due to

when and how the radiative correction was implemented: the correction for RSN2010 is
about 0.1 K warmer than those from RSN2005. To identify the possible RS92
temperature biases due to changes of radiation correction table (i.e., RSN2010), we





compare mean $\Delta T$ (RS92) from January 2007 to December 2010 (i.e., $\Delta T$ (RS92$_{200701\text{-}}$
$_{201012}$) to those from January 2011 to April 2014. Results show that there are no obvious
$\Delta T$ (RS92) change between these two periods for the RS92 sondes over United States and
Germany in 20hPa. However, the daytime temperature difference between $\Delta T$
(RS92$_{200701\text{-}201012}$) and $\Delta T$ (RS92$_{201101\text{-}201404}$) over Australia, Canada, England, and Brazil
show obvious close to 0.1 K to 0.15 K difference varying at different heights. Changing
sensors independently of the appropriate radiation correction introduces extra
uncertainties of the RS92 trends. The relatively small temperature difference between
these two periods over the United States is most likely a statistical artifact due to the very
small number of coincidences in this period. The relatively small temperature difference
between these two periods over the Germany may because the DWD implemented the
updated radiation correction for the Vaisala RS92 not in 2011, but in the spring of 2015,
to avoid inconsistencies with corrections already implemented in their data assimilation
system. This also indicates the importance of establishing traceability through careful
documentation and metadata tracking, which is especially crucial for radiosonde data
used in climate studies.

4. We used time series of RAOB-RO anomalies to indicate the long term stability

for each individual sonde types. The uncertainties are from the combined effects of i)
uncorrected solar zenith angle dependent biases, ii) change of radiation correction, iii)
when and how the radiation correction was implemented, and iv) small samples used in
the time series and trend analysis. Results show that the time series of $\Delta T^{Time}$ (RS92) at all
regions are, in general, persistent in time with a small difference during the daytime and
nighttime in each individual regions. Other sensors have much larger variation than those



of Vaisala RS92. While $\Delta_{Sippican}^{Time}$ and $\Delta_{Shanghai}^{Time}$ are largely constant in time, $\Delta_{VIZ-B2}^{Time}$ has
obvious seasonal variations with a negative trend and night and positive trend during
daytime. $\Delta_{AVK}^{Time}$ has an irregular seasonal variation particularly during daytime, with a
large warm bias.

5. We found that the variation of mean $\Delta T^{Time}$ at different regions is highly related to

the corresponding variation of SZA especially for VIZ-B2 and AVK-MRZ during the
daytime where the Sippican MARK IIA over USA and Shanghai $\Delta T^{Time}$ do not show
significant seasonal variation. The daytime trend of anomalies for RAOB and RO
temperature at 50 hPa for United States and Germany are equal to (0.00, -0.02) K/5yrs,
the trend of anomalies are equal to (0.18, 0.24, 0.26, 0.12) K/5yrs over Australia, Canada,
England, and Brazil, respectively. The trend anomaly between RS92 and RO from June
2006 to April 2014 is within +/-0.09 K/5yrs globally. The trend anomaly between RS80
and RO is about 0.19 K/5yrs during the day and 0.11 K/5yrs during the night. The
daytime temperature trend anomalies for $\Delta T_{MARK-IIA}^{Deseason}$, $\Delta T_{VIZ-B2}^{Deseason}$, $\Delta T_{AVK}^{Deseason}$, and $\Delta T_{Shanghai}^{Deseason}$
are (0.40, 0.47, -0.14, 0.18) K/5yrs, which are much larger than those of RS92.

Note that the analyses we performed here do not include other error sources (i.e.,

cloud radiative effect, ventilation, and sensor orientation, meta data errors) mentioned by
Dirksen et al., (2014). Since it is not possible to investigate these errors, we assume these
errors introduce more or less random errors when a relative large sample is used. In
addition, although RO derived temperature data are not directly traceable to the
international standard of units (SI traceability), it has been shown that the high precision
nature does preserve through the inversion procedures (Ho et al., 2009a, 2011). This
makes RO derived temperature uniquely useful for assessing the radiosonde temperature



biases and their long-term stability including the seasonal and inter-annual variability in
the lower stratosphere. Results from this study also demonstrate the potential usage of
RO data to identify RAOB temperature biases for different sensor types.

**Acknowledgments**

This work is supported by the NSF CAS AGS-1033112. The authors

acknowledge the contributions to this work from members of the COSMIC team at
UCAR.






















**Appendix A: The Quality of GPS RO Data as Benchmark References and the New**
**Reprocessed Package**

**A1. The Quality of GPS RO Data as Benchmark References for Climate Studies**

While the position and speed of the GPS and low earth orbit (LEO) satellites are

known, we can inverse the time delay to bending angles, refractivity, and temperature
vertical distribution with high precision and accuracy (Ho et al., 2009a,b, 2012). While
time delay and bending angle are traceable to the international standard of units (SI
traceability), the derived temperature profiles are not. To investigate the structural
uncertainty of RO temperature profiles, Ho et al., (2009a and 2011) compared CHAMP
(CHAllenging Minisatellite Payload) temperature profiles generated from multiple
centers when different inversion procedures were implemented. Results shown that the
mean RO temperature biases for one center (for example from UCAR) relative to the all
center mean is within ±0.1K from 8 km to 30 km except for south pole above 25 km (see
Fig. 6d in Ho et al., 2011). Ho et al., (2007, 2009b) demonstrated that the RO derived
temperature profiles in the lower stratosphere are extremely useful to identify and
calibrate the inter-satellite microwave brightness temperature differences form Advanced
Microwave Sounding Units (AMSU) and Microwave Sounding Units (MSU) on board
different satellite missions.  In this study, UCAR RO temperature profiles will be used in
this study.

GPS RO observations are of high vertical resolution (from ~60 m near the surface

to ~1.5 km at 40 km). The mean temperature difference between the collocated soundings
of COSMIC and CHAMP is within 0.1 K from 200 hPa to 20 hPa (Ho et al., 2009b;



Anthes et al., 2008; Foelsche et al., 2009). Schreiner et al., (2014) compared re-processed
COSMIC and Metop-A/GRAS bending angles produced at CDAAC. The mean COSMIC
and Metop-A/GRAS bending angle differences are about 0.02–0.03 µrad which
demonstrates the reproducibility of COSMIC and Metop-A/GRAS. The mean layer
temperature difference between 200 hPa to 10 hPa is within 0.05 K (not shown). This is
consistent with those between COSMIC and CHAMP at the same height (Ho et al.,
2009a). The precision of RO temperature is ~ 0.1 K (e.g., Anthes et al., 2008; Ho et al.,
2009a), and the precision of the trend of RO derived temperature data is within ±0.06
K/5yrs (Ho et al., 2012). To estimate the accuracy of RO temperature in the upper
troposphere and lower stratosphere, Ho et al., (2010) compared RO temperature from 200
hPa to 10 hPa to those from Vaisala-RS92 in 2007 where more than 10,000 pairs of
Vaisala-RS92 and COSMIC coincident data are collected. The mean bias in this height
range is equal to -0.01 K with a mean standard deviation of 2.09K. Although the quality
of Vaisala-RS92 may vary in different regions (see Section 4.1), this comparison
demonstrates the quality of RO temperature profiles in this height range.

**A2. Brief Description of the New Inversion Package from CDAAC**

Comparing with the previous version, the new inversion package used improved

precise orbit determination (POD) and excess phase processing algorithm, where a high-
precision, multiple Global Navigation Satellite System (GNSS) data processing
software (i.e., Bernese Version 5.2, Dach et al., (2015)) is applied for clock estimation
and time transfer. In the reprocessing package, the POD for COSMIC and Metop-
A/GRAS are implemented separately (Schrein et al., 2011). Compared to the real-time
processed RO data, much improved and more completed satellite POD data are used in



the reprocessed package. The re-processed COSMIC and Metop-A/GRAS data would
produce more consistent and accurate RO variables than those from post-processed
(periodically updated inversion packages were used) and real-time processed datasets.






















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





## Figure Captions

Figure 1. Global distribution of radiosonde stations colored by radiosonde types. Radiosonde types updated from June 2006 to December 2015 are used. The percentage of each type of radiosonde used among all stations is listed. For those stations that radiosonde types are changed during this period, the latest updated radiosonde type is used in this plot. Vaisala RS92 ship observations contain less than 3% of the total RS92 profiles.

Figure 2. Mean RAOB-RO temperature biases at 50 hPa for the RAOB-RO ensembles from June 2006 to December 2015 for a) daytime, and b) nighttime. Only those stations containing more than 50 RO-ROAB pairs are plotted.

Figure 3. Comparisons of temperature between RS92 and RO for daytime over a) United States, b) Australia, c) Germany, d) Canada, e) England, and f) Brazil. The red line is the mean difference; the black line is the standard deviation of the mean difference; the dotted line is the sample number. The top X axis shows the sample number. The same symbols are also used for the following plots.

Figure 4. Comparisons of temperature between RS92 and RO for nighttime over a) United States, b) Australia, c) Germany, d) Canada, e) England, and f) Brazil.

Figure 5. The mean temperature biases (RS92 minus RO) at 50 hPa varying for SZA from 0 degrees to 180 degrees for a) United States, b) Australia, c) Germany, d) Canada, e) England, and f) Brazil. The red cross is the mean difference for each 5 SZA bins; the red vertical line is the standard deviation of error defined as standard deviation divided by sample numbers; the vertical red lines superimposed on the mean are the standard error of the mean; the black line to indicate zero mean; the blue dash line is the sample number. The right Y axis shows the sample number. Only bins for more than 50 RAOB-RO pairs are plotted.

Figure 6. Comparisons of temperature between radiosonde and RO during the daytime for a) Sippican over United States minus RO, b) VIZ-B2 over United States minus RO, c) Russian Sonde minus RO, d) Shanghai minus RO.

Figure 7. Comparisons of temperature between radiosonde and RO during the nighttime for a) Sippican over United States minus RO, b) VIZ-B2 over United States minus RO, c) Russian Sonde minus RO, d) Shanghai minus RO.

Figure 8. The mean temperature biases at 50 hPa varying for SZA from 0 degrees to 180 degrees for a) Sippican over United States minus RO, b) VIZ-B2 over United States minus RO, c) Russian Sonde minus RO, d) Shanghai minus RO. Only bins for more than 50 RAOB-RO pairs are plotted.

Figure 9. The temperature differences between RS92 – RO from January 2007 to December 2010 ($\Delta T$ (RS92$_{200701-201012}$) and those from January 2011 to December 2015 ($\Delta T$ (RS92$_{201101-201512}$) over a) United States, b) Australia, c) Germany, d) Canada, e) England, and f) Brazil.

Figure 10. The time series of monthly mean temperature differences to RO at 50 hPa for RS92 for a) United States, b) Australia, c) Germany, d) Canada, e) England, and f) Brazil. The red cross is the mean difference for RS92 minus RO temperature at 50 hPa during the daytime and the blue cross is for that during the nighttime; the vertical lines superimposed on the mean values are the standard error of the mean for daytime and nighttime, respectively; the back line indicates zero temperature bias; the pink/green dash line is the sample number for the daytime and nighttime, respectively. The right Y axis shows the sample number. The same symbols are also used for the following plots.





Figure 11. The time series of temperature anomaly in 50 hPa for a) Sippican over United States minus
RO, b) VIZ-B2 over United States minus RO, c) Russian Sonde minus RO, d) Shanghai minus RO.
Figure 12. The time series of de-seasonalized temperature anomaly in 50 hPa for RS92 for a) United
States, b) Australia, c) Germany, d) Canada, e) England, and f) Brazil. The 95% confidence intervals
for slopes are shown in the parentheses.
Figure 13. The time series of de-seasonalized temperature anomaly in 50 hPa for a) Sippican over
United States minus RO, b) VIZ-B2 over United States minus RO, c) Russian Sonde minus RO, d)
Shanghai minus RO. The 95% confidence intervals for slopes are shown in the parentheses.





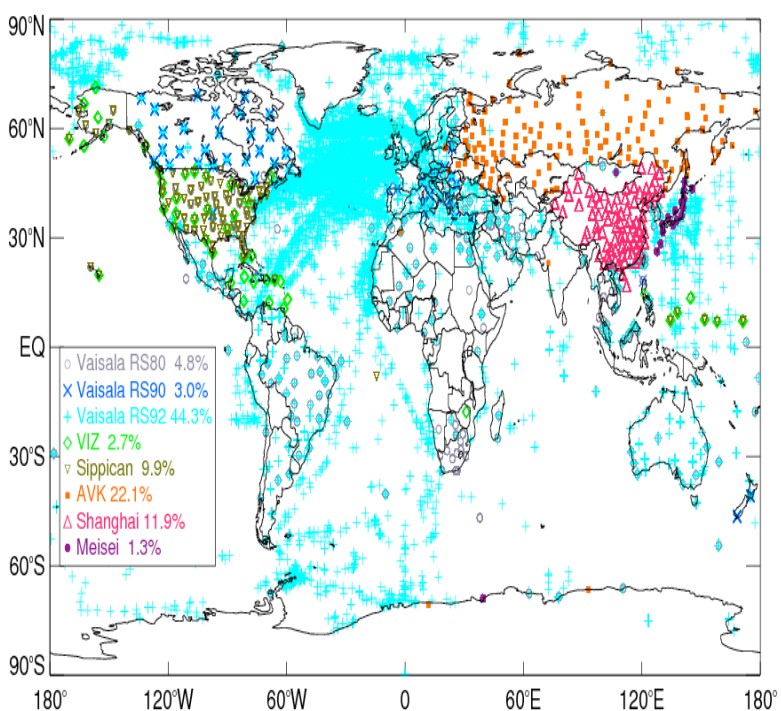

Figure 1. Global distribution of radiosonde stations colored by radiosonde types. Radiosonde types updated from June 2006 to December 2015 are used. The percentage of each type of radiosonde used among all stations is listed. For those stations that radiosonde types are changed during this period, the latest updated radiosonde type is used in this plot. Vaisala RS92 ship observations contain less than 3% of the total RS92 profiles.




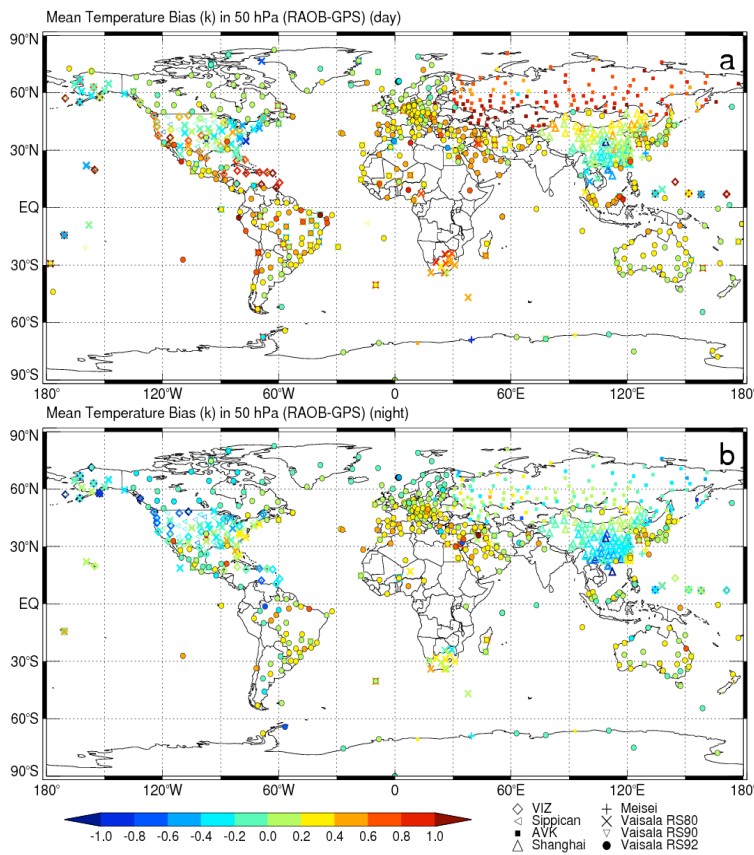

Figure 2. Mean RAOB-RO temperature biases at 50 hPa for the RAOB-RO ensembles from June
2006 to December 2015 for a) daytime, and b) nighttime. Only those stations containing more than 50
RO-ROAB pairs are plotted.




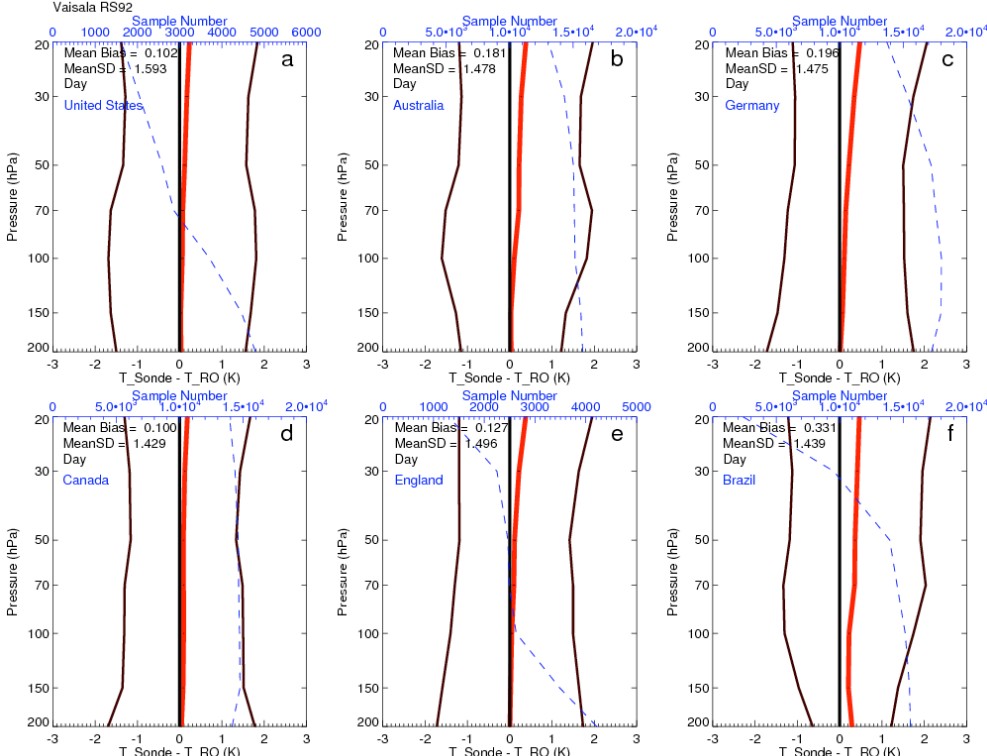

Figure 3. Comparisons of temperature between RS92 and RO for daytime over a)
United States, b) Australia, c) Germany, d) Canada, e) England, and f) Brazil. The red
line is the mean difference; the black line is the standard deviation of the mean difference;
the horizontal black lines superimposed on the mean are the standard error of the mean;
the dotted line is the sample number. The top X axis shows the sample number. The same
symbols are also used for the following plots.






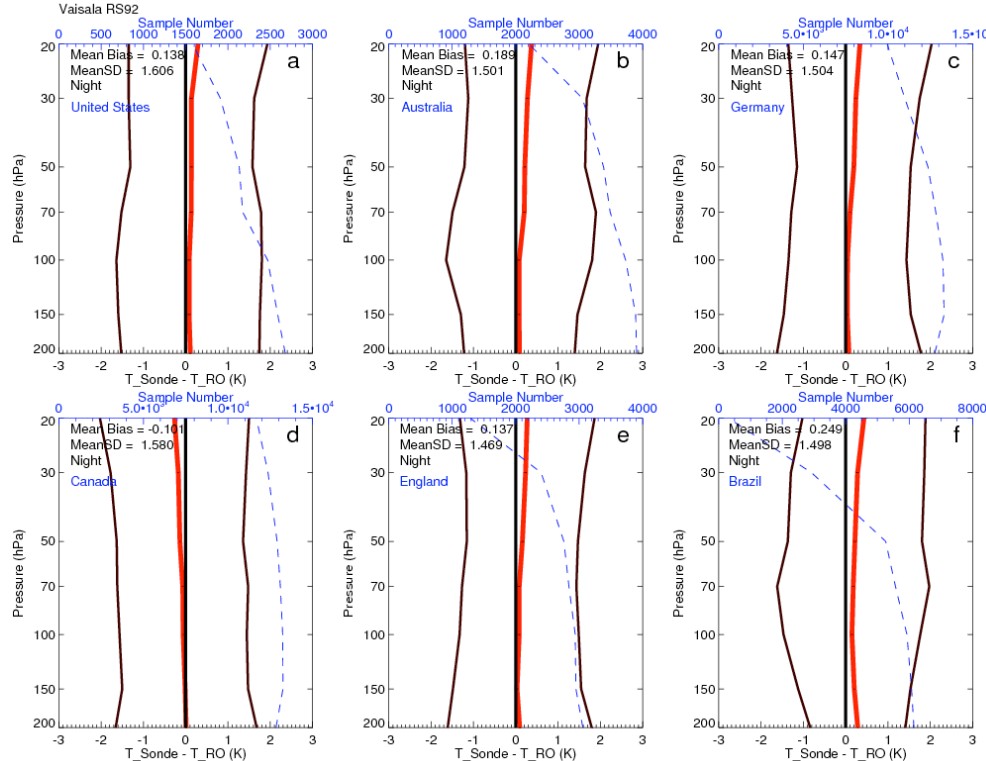

Figure 4. Comparisons of temperature between RS92 and RO for nighttime over a)
United States, b) Australia, c) Germany, d) Canada, e) England, and f) Brazil.





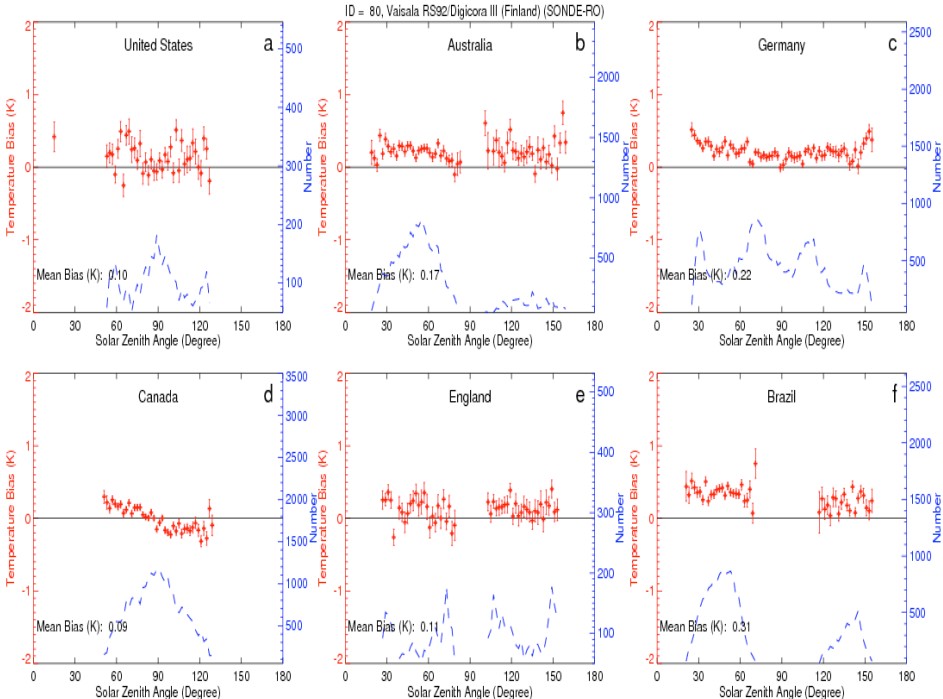

Figure 5.  The mean temperature biases (RS92 minus RO) at 50 hPa varying for SZA from 0 degrees
to 180 degrees for a) United States, b) Australia, c) Germany, d) Canada, e) England, and f) Brazil.
The red cross is the mean difference for each 5 SZA bins; the red vertical line is the standard deviation
of error defined as standard deviation divided by sample numbers; the vertical red lines superimposed
on the mean are the standard error of the mean; the black line to indicate zero mean; the blue dash line
is the sample number. The right Y axis shows the sample number. Only bins for more than 50 RAOB-
RO pairs are plotted.




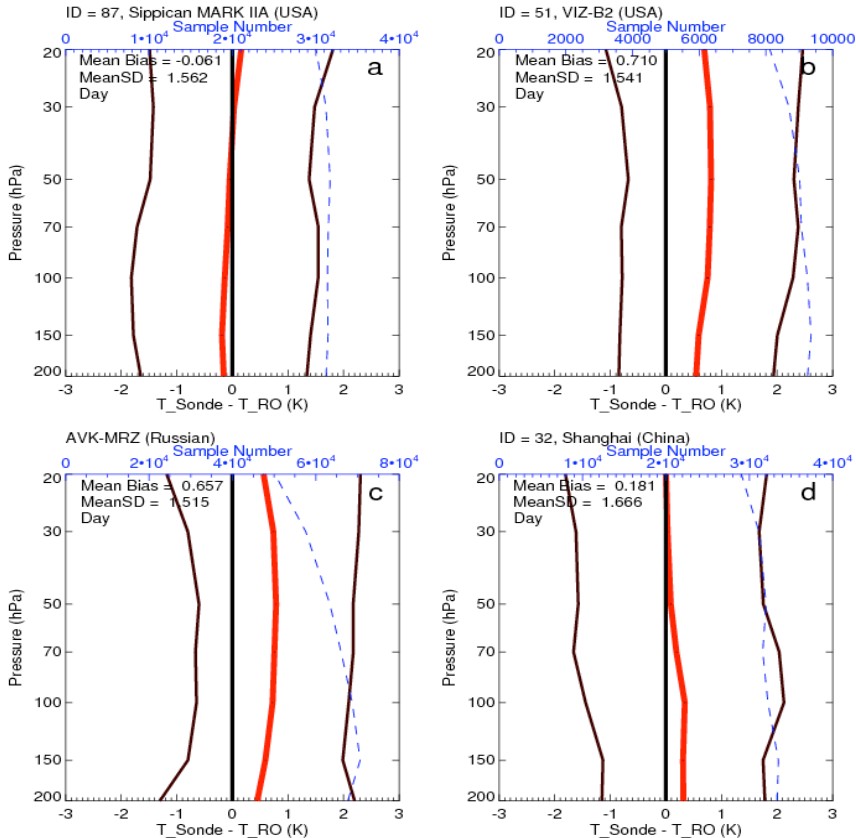

Figure 6. Comparisons of temperature between radiosonde and RO during the daytime for a) Sippican
over United States minus RO, b) VIZ-B2 over United States minus RO, c) Russian Sonde minus RO,
d) Shanghai minus RO.





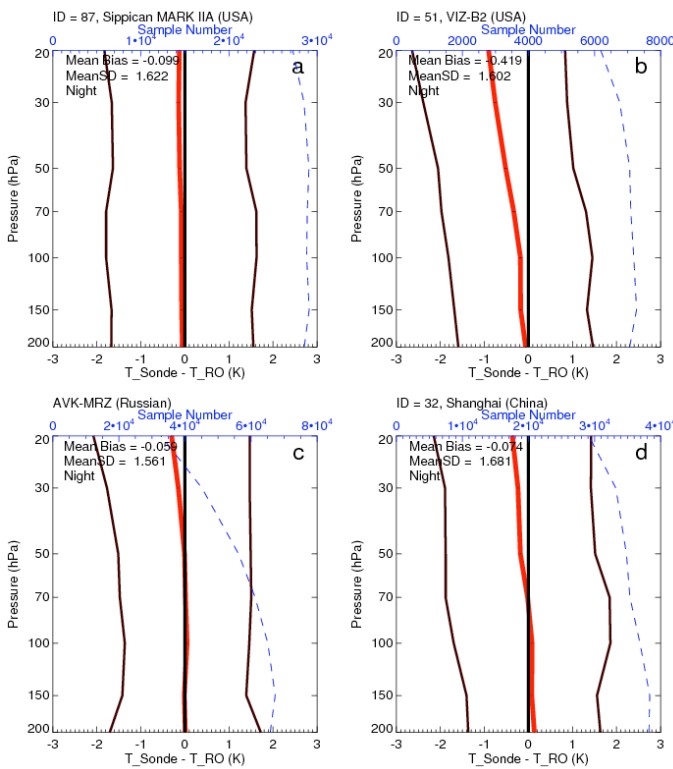

Figure 7. Comparisons of temperature between radiosonde and RO during the nighttime for a)
Sippican over United States minus RO, b) VIZ-B2 over United States minus RO, c) Russian Sonde
minus RO, d) Shanghai minus RO.




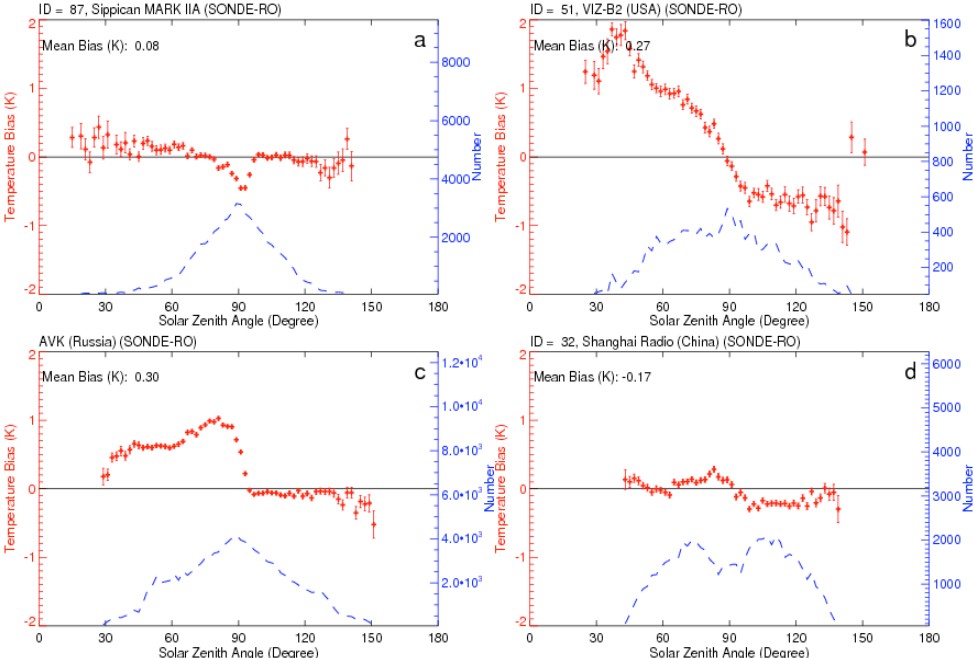

Figure 8.  The mean temperature biases at 50 hPa varying for SZA from 0 degrees to 180 degrees for a)
Sippican over United States minus RO, b) VIZ-B2 over United States minus RO, c) Russian Sonde
minus RO, d) Shanghai minus RO.  Only bins for more than 50 RAOB-RO pairs are plotted.



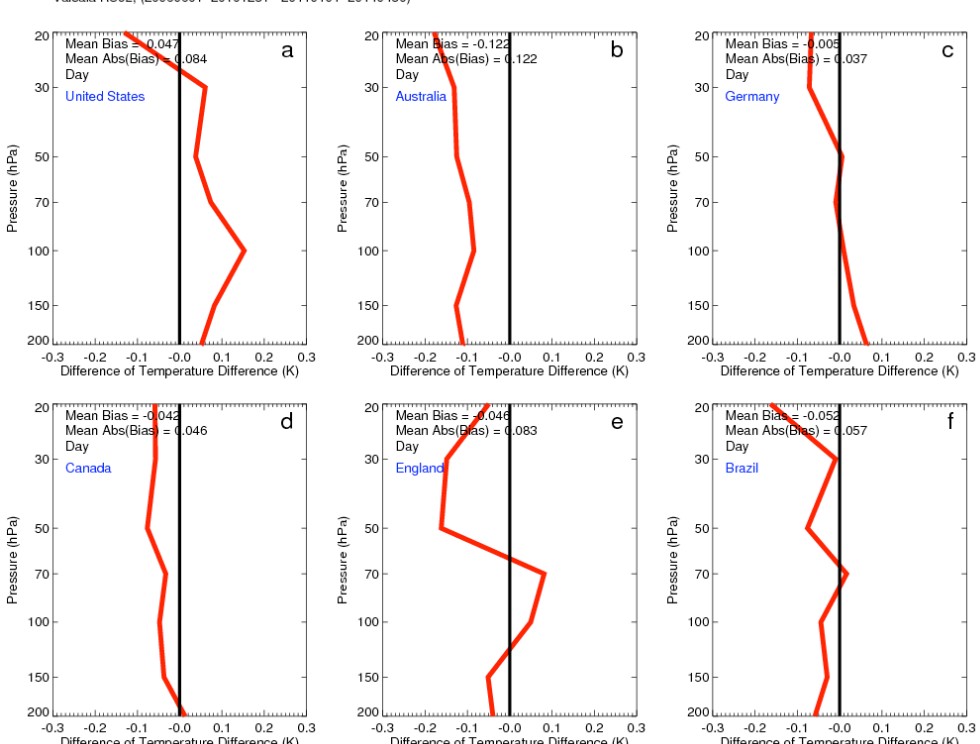

Figure 9. The temperature differences between RS92 – RO from January 2007 to December 2010 ($\Delta T$
(RS92$_{200701\text{-}201012}$) and those from January 2011 to December 2015 ($\Delta T$ (RS92$_{201101\text{-}201512}$) over a)
United States, b) Australia, c) Germany, d) Canada, e) England, and f) Brazil.






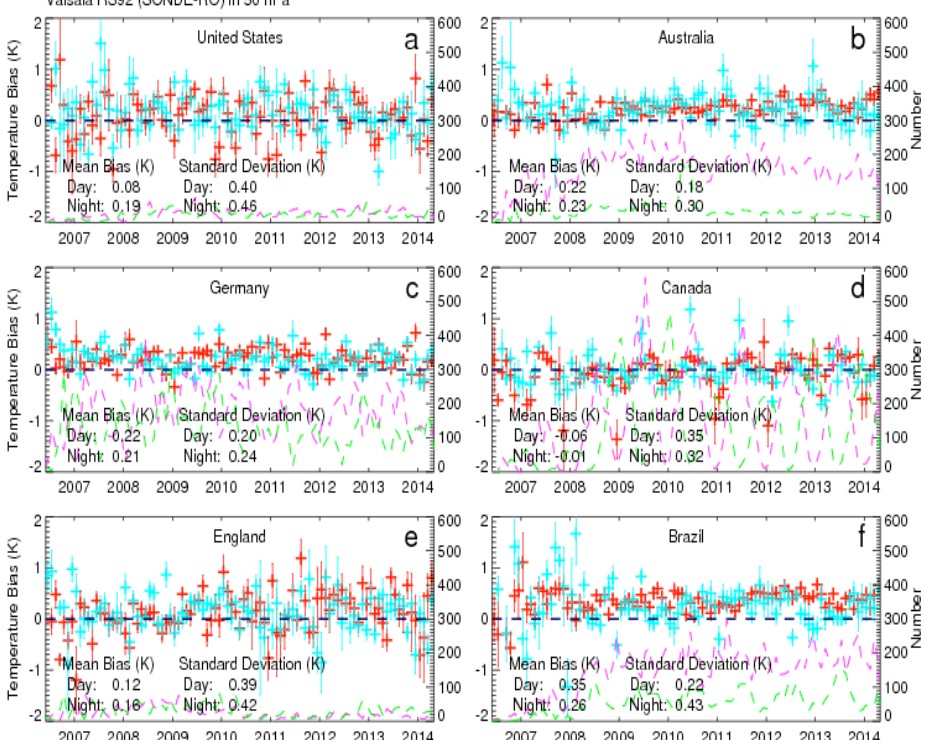

Figure 10. The time series of monthly mean temperature differences to RO at 50 hPa for RS92 for a)
United States, b) Australia, c) Germany, d) Canada, e) England, and f) Brazil. The red cross is the
mean difference for RS92 minus RO temperature at 50 hPa during the daytime and the blue cross is
for that during the nighttime; the vertical lines superimposed on the mean values are the standard error
of the mean for daytime and nighttime, respectively; the back line indicates zero temperature bias; the
pink/green dash line is the sample number for the daytime and nighttime, respectively. The right Y
axis shows the sample number. The same symbols are also used for the following plots.





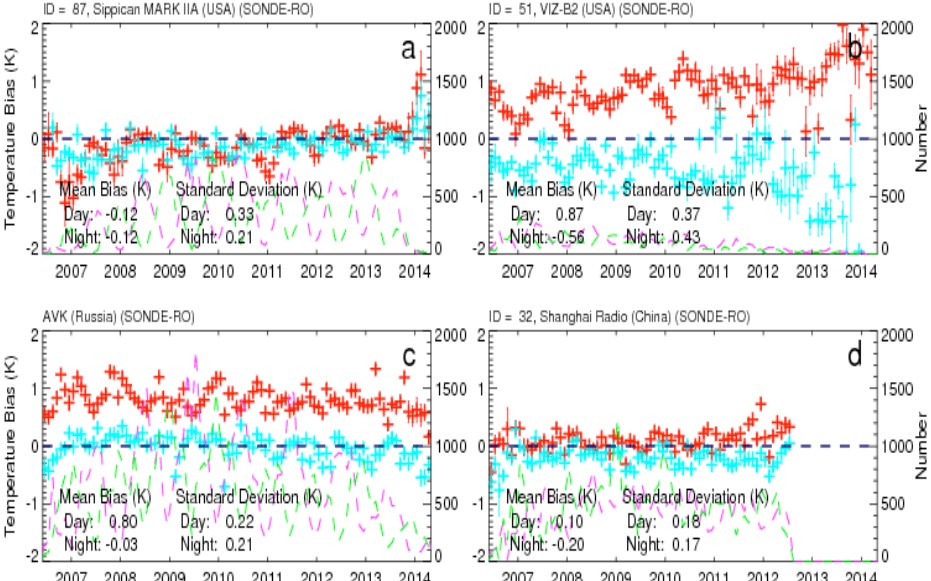

Figure 11. The time series of temperature anomaly in 50 hPa for a) Sippican over United States minus RO, b) VIZ-B2 over United States minus RO, c) Russian Sonde minus RO, d) Shanghai minus RO.






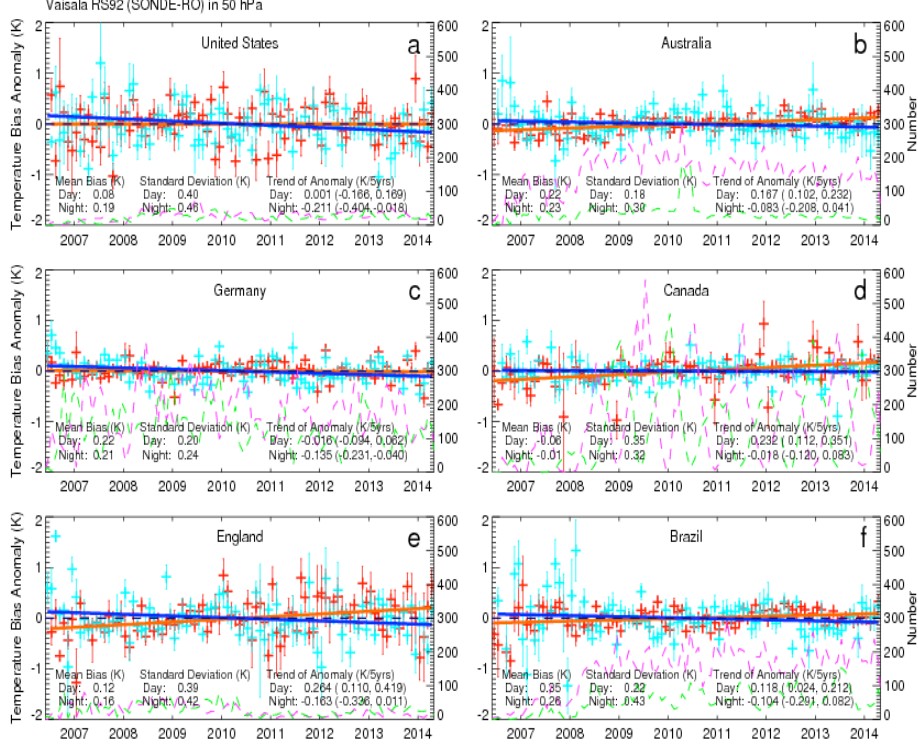

Figure 12. The time series of de-seasonalized temperature anomaly in 50 hPa for RS92 for a) United
States, b) Australia, c) Germany, d) Canada, e) England, and f) Brazil. The 95% confidence intervals
for slopes are listed in the parentheses.

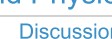




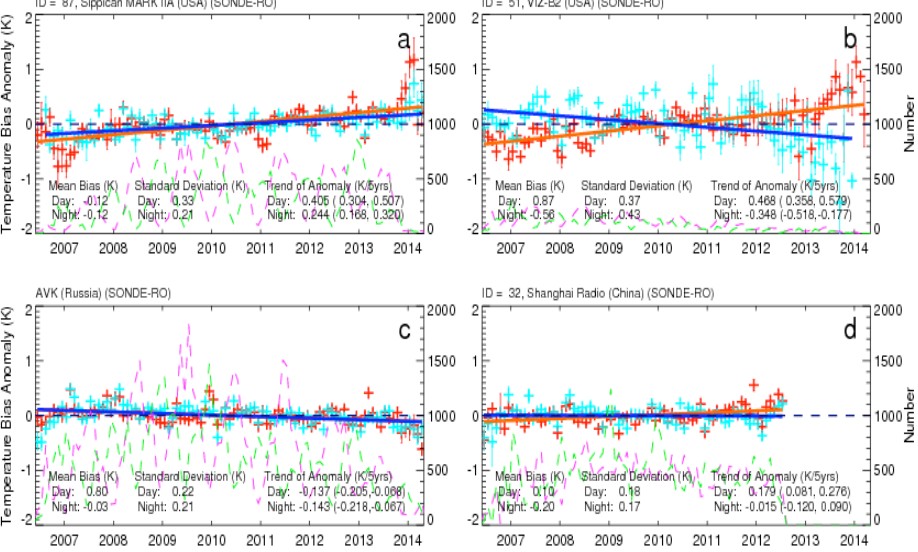

Figure 13. The time series of de-seasonalized temperature anomaly in 50 hPa for a) Sippican over
United States minus RO, b) VIZ-B2 over United States minus RO, c) Russian Sonde minus RO, d)
Shanghai minus RO. The 95% confidence intervals for slopes are listed in the parentheses.





Table 1. Summary of the availability for different instrument types and their solar absorptivity (α) and
sensor infrared emissivity (ε) for the corresponding thermocap and thermistor and the sample number
of RAOB-RO pairs used in this study from June 2006 to December 2015.

| | ID | Sensor type | Availability | Solar absorptivity | Infrared emissivity | Number of RO-RAOB pairs |
|---|---|---|---|---|---|---|
| RS80 | 37 | Bead thermocap | 1981~ 2014 | 0.15[*Luers and Eskridge*, 1998] | 0.02 | 1624 |
| Vaisala RS80-57H | 52 | Bead thermocap | early 1990s [*Redder et al.*, 2004] ~ Jul 2012 | 0.15 | 0.02 | 13192 |
| Vaisala RS80/Loran | 61 | Bead thermocap | ~ 2014 | 0.15 | 0.02 | 11591 |
| Vaisala RS80/Digicora III | 67 | Bead thermocap | ~ 2012 | 0.15 | 0.02 | 2864 |
| Vaisala RS90/Digicorn I, II | 71 | Thin wire F-thermocap [*Sun et al.*, 2010] | 1995 ~ 2014 | 0.15[*Luers*, 1997] | 0.02 | 18082 |
| Vaisala RS92/Digicora I/II | 79 | Thin wire F-thermocap [*Sun et al.*, 2010] | 2003 ~ 2014 | 0.15 | 0.02 | 40478 |
| Vaisala RS92/Digicora III | 80 | Thin wire F-thermocap | 2004~2014 | 0.15 | 0.02 | 184542 |
| Vaisala RS92/Autosonde | 81 | Thin wire F-thermocap | 2011~2014 | 0.15 | 0.02 | 42577 |
| AVK-MRZ | 27 | Rod thermistor [*Sun et al.*, 2010] | ~ 2014 | 0.2[*He et al.*, 2009] | 0.04 | 48954 |
| AVK-BAR Russian | 58 | Rod thermistor | 2007 ~ 2014 | 0.2 | 0.04 | 26020 |
| AVK-MRZ (Russian) | 75 | Rod thermistor | ~ 2013 | 0.2 | 0.04 | 9472 |
| MARL-A or Vektor-M-MRZ (Russian) | 88 | Rod thermistor | ~ 2014 | 0.2 | 0.04 | 23326 |
| MARL-A or Veltor-M-BAR (Russian) | 89 | Rod thermistor | ~ 2014 | 0.2 | 0.04 | 25715 |
| VIZ-B2 | 51 | Rod thermistor[*Sun et al.*, 2010] | 1997[*Elliott et al.*, 2002]~ 2014 | 0.15[*Luers and Eskridge*, 1998] | 0.86 | 16310 |
| Sippican MARK II A Chip | 87 | Chip thermistor[*Sun et al.*, 2010] | 1998[*Elliott et al.*, 2002]~ 2014 | 0.07[*Luers and Eskridge*, 1998] | 0.85 | 59775 |
| Shanghai | 32 | Rod thermistor | 1998 ~ 2012 | <0.07 [*Wei*, 2011] | >0.90 | 71605 |
| Meisei Japan | 47 | Thermistor [*KOBAYASHI et al.*, 2012] | 1994 ~ 2013 | 0.18[*Luers and Eskridge*, 1998] | 0.84 | 7888 |






Table 2.  Mean and standard deviation of temperature differences (K) from the layer from 200 hPa to
20 hPa between RO and eight types of radiosonde[a,b]. [a]The values of standard deviations of temperature
differences are shown in the parentheses. [b]The sample number are for the ROAB-RO pairs available
in the same time period.

| | ID | All Day and night mean(std)/ sample numbers | Day mean(std)/ sample numbers | Night mean(std)/ sample numbers |
|---|---|---|---|---|
| Vaisala RS80 | 37, 52, 61, 67 | 0.10 (1.54)/29271 | 0.10 (1.53)/15947 | 0.09 (1.55)/13324 |
| Vaisala RS90 | 71 | 0.13 (1.54)/18082 | 0.16 (1.51)/8758 | 0.11 (1.57)/9324 |
| Vaisala RS92 | 79, 80, 81 | 0.16 (1.52)/267597 | 0.20 (1.50)/161019 | 0.09 (1.55)/106578 |
| AVK | 27, 75, 88, 89, 58 | 0.33 (1.58)/133487 | 0.66 (1.51)/67679 | -0.06 (1.56)/65808 |
| VIZ-B2 | 51 | 0.22 (1.67)/16310 | 0.71 (1.54)/9246 | -0.42 (1.60)/7064 |
| Sippican MARKIIA Chip | 87 | -0.08 (1.59)/59775 | -0.06 (1.56)/31230 | -0.10 (1.62)/28545 |
| Shanghai | 32 | 0.05 (1.68)/71605 | 0.18 (1.67)/33360 | -0.07 (1.68)/38245 |
| Meisei Japan | 47 | 0.11 (1.69)/7888 | 0.03 (1.71)/3849 | 0.19 (1.66)/4039 |







Table 3. Mean, standard deviation (std) of monthly temperature differences (K), trend of temperature
anomaly (K/5yrs), and root mean square (RMS) of RS92-RO time series at 50 hPa over United States,
Australia, Germany, Canada, England, and Brazil.

|  | United States |  | Australia |  | Germany |  | Canada |  | England |  | Brazil |  |
| --- | --- | --- | --- | --- | --- | --- | --- | --- | --- | --- | --- | --- |
|  | Day | Night | Day | Night | Day | Night | Day | Night | Day | Night | Day | Night |
| Mean Bias | 0.08 | 0.19 | 0.22 | 0.23 | 0.22 | 0.21 | -0.06 | -0.01 | 0.12 | 0.16 | 0.35 | 0.26 |
| std of Mean Bias | 0.4 | 0.46 | 0.18 | 0.3 | 0.2 | 0.24 | 0.35 | 0.32 | 0.39 | 0.42 | 0.22 | 0.43 |
| Trend of Anomaly (K/ 5 yrs) | 0.001 | -0.211 | 0.167 | -0.083 | -0.016 | -0.135 | 0.232 | -0.018 | 0.264 | -0.163 | 0.118 | -0.104 |
| Trend of RO Temperature (K/5yrs) | 0.941 | 0.506 | -0.26 | 0.082 | 0.29 | 0.708 | -0.69 | -0.534 | 0.509 | 1.143 | -0.076 | -0.354 |
| RMS of ANOM | 0.365 | 0.439 | 0.161 | 0.275 | 0.173 | 0.22 | 0.276 | 0.215 | 0.358 | 0.392 | 0.212 | 0.398 |






Table 4. Mean, standard deviation (std) of monthly temperature differences (K), trend of temperature
anomaly (K/5yrs), and root mean square (RMS) of RS92-RO time series at 150 hPa over United States,
Australia, Germany, Canada, England, and Brazil.

| | United States | | Australia | | Germany | | Canada | | England | | Brazil | |
|---|---|---|---|---|---|---|---|---|---|---|---|---|
| | Day | Night | Day | Night | Day | Night | Day | Night | Day | Night | Day | Night |
| Mean Bias | 0 | 0.09 | 0.03 | 0.08 | 0.07 | 0.06 | 0.03 | 0.12 | 0.04 | 0.05 | 0.21 | 0.23 |
| STD of Monthly Mean Bias | 0.33 | 0.35 | 0.13 | 0.22 | 0.2 | 0.25 | 0.35 | 0.52 | 0.37 | 0.33 | 0.16 | 0.27 |
| Trend of ANOM (K/ 5 yrs) | -0.134 | -0.228 | 0.117 | -0.072 | -0.02 | -0.189 | 0.226 | -0.21 | 0.056 | -0.083 | 0.004 | -0.014 |
| Trend of RO Temperature (K/5yrs) | 1.508 | 1.134 | -0.2 | -0.232 | 0.428 | 0.717 | -0.797 | 0.217 | 0.562 | 1.011 | 0.085 | 0.439 |
| RMS of ANOM | 0.302 | 0.328 | 0.126 | 0.197 | 0.18 | 0.222 | 0.301 | 0.466 | 0.346 | 0.305 | 0.141 | 0.242 |





Table 5.  Mean, standard deviation (std), trend (K/5yrs), and root mean square (RMS) of time series of
temperature anomaly at 50 hPa for global Vaisala (RS80, RS90, and RS92), and other sensor types in
the North hemisphere mid-latitude (60°N-20°N). The 95% confidence intervals for trend of anomaly
are listed in the parentheses.

|  | ID | Day | | | | Night | | | |
|---|---|---|---|---|---|---|---|---|---|
|  |  | Mean Bias | STD Of MB | Trend of ANOM (k/5yrs) | RMS of ANOM | Mean Bias | STD Of MB | Trend of ANOM (k/5yrs) | RMS of ANOM |
| RS80 | 37,52, 61,67 | 0.18 | 0.29 | 0.187 (0.073,0.301) | 0.268 | 0.13 | 0.33 | 0.114(-0.019,0.248) | 0.301 |
| RS90 | 71 | 0.16 | 0.29 | -0.006 (-0.123,0.111) | 0.26 | 0.17 | 0.38 | 0.043(-0.115,0.201) | 0.352 |
| RS92 | 79,80, 81 | 0.22 | 0.07 | 0.074 (0.051,0.097) | 0.062 | 0.12 | 0.12 | -0.094(-0.131,-0.057) | 0.093 |
| Russia | 27,75, 88,89 58 | 0.8 | 0.22 | -0.137 (-0.205,-0.068) | 0.164 | -0.03 | 0.21 | -0.143(-0.218,-0.067) | 0.18 |
| VIZ-B2 | 51 | 0.87 | 0.37 | 0.468 (0.358,0.579) | 0.322 | -0.56 | 0.43 | -0.348(-0.518,-0.177) | 0.386 |
| Sippican MARKIIA Chip | 87 | -0.12 | 0.33 | 0.405 (0.304,0.507) | 0.292 | -0.12 | 0.21 | 0.244(0.168,0.320) | 0.197 |
| Shanghai | 32 | 0.1 | 0.18 | 0.179 (0.081,0.276) | 0.161 | -0.2 | 0.17 | -0.015(-0.120,0.090) | 0.159 |
| Meisei Japan | 47 | 0.07 | 0.69 | 0.006 (-0.353,0.365) | 0.619 | 0.05 | 0.51 | -0.086(-0.369,0.197) | 0.494 |




Table 6. Mean, standard deviation (std), trend (K/5yrs), and root mean square (RMS) of time series of
temperature anomaly at 150 hPa for global Vaisala (RS80, RS90, and RS92), and other sensor types in
the North hemisphere mid-latitude (60°N-20°N). The 95% confidence intervals for trend of anomaly
are shown in the parentheses.

| | | Day | | | | Night | | | |
|---|---|---|---|---|---|---|---|---|---|
| | | Mean Bias | STD Of MB | Trend of ANOM (k/5yrs) | RMS of ANOM | Mean Bias | STD Of MB | Trend of ANOM (k/5yrs) | RMS of ANOM |
| RS80 | 37,52, 61,67 | 0.18 | 0.19 | 0.045 (-0.036,0.126) | 0.18 | 0.21 | 0.29 | 0.063(-0.055,0.181) | 0.263 |
| RS90 | 71 | 0.1 | 0.31 | -0.058 (-0.181,0.065) | 0.275 | 0.11 | 0.33 | -0.065(-0.203,0.072) | 0.307 |
| RS92 | 79,80, 81 | 0.08 | 0.05 | 0.013 (-0.005,0.031) | 0.041 | 0.05 | 0.08 | -0.068(-0.094,-0.042) | 0.066 |
| Russia | 27,75, 88,89 58 | 0.54 | 0.19 | -0.194 (-0.254,-0.134) | 0.16 | 0 | 0.21 | -0.147(-0.199,-0.094) | 0.135 |
| VIZ-B2 | 51 | 0.65 | 0.38 | 0.370 (0.227,0.514) | 0.362 | -0.15 | 0.3 | -0.051(-0.182,0.079) | 0.272 |
| Sippican MARKIIA Chip | 87 | -0.23 | 0.19 | 0.217 (0.148,0.285) | 0.181 | -0.1 | 0.18 | 0.160(0.095,0.226) | 0.158 |
| Shanghai | 32 | 0.32 | 0.12 | -0.086 (-0.149,-0.023) | 0.1 | 0.08 | 0.19 | -0.302(-0.394,-0.210) | 0.176 |
| Meisei Japan | 47 | 0.06 | 0.53 | -0.102 (-0.371,0.168) | 0.458 | 0.19 | 0.5 | 0.001(-0.271,0.272) | 0.455 |
