# Peer review of "Characterization of the Long-term Radiosonde Temperature Biases in"

_Atmospheric Chemistry and Physics, 2016_

## Referee Comment (RC1) · R. Anthes (Referee) · 30 Oct 2016

The authors have done a lot of careful work and there are interesting results in their paper on the different accuracies, stabilities and trends of various types of radiosonde data. The authors use radio occultation (RO) data in the upper troposphere and lower stratosphere as a reference data set for the period June 2006 to April 2014. Since radiosonde data are often used in climate studies, it is important to document the accuracies and uncertainties of different types of radiosondes in different countries.

However, the paper is too long (58 pages) and Sections 3-5 on results are tedious and

difficult to read because of too much detail in the text that merely repeats what is in the tables and figures, as well as too many symbols in the text (e.g. $\Delta T$ (RS92200701-201012)). Furthermore, there is too much listing and discussion of statistics that are small and probably not significant or of general interest. The reader is thus overwhelmed with the reporting of many numbers without a focus on what is really important. The paper would be improved and have much more impact if it were shortened significantly and only the major results included in the text.

There are many statistics of radiosonde minus RO temperatures from various types of radiosondes at different levels of the atmosphere between 200 and 20 hPa over six different regions of the world. It is not clear which of these results are statistically significant. This makes interpretation of the results difficult as we could be looking at small, quasi-random differences that have no physical meaning, nor even meaning relative to the specific types of radiosonde data. Differences are often 0.1K or less, which are well below the accuracy of radiosonde sensors. When the different atmospheric sampling volumes of the radiosondes and RO are considered, sampling errors alone can be much larger than 0.1K.

The authors compute trends of the differences between individual types of radiosonde and RO over a 7-year period. Most of the trends are small (of order 0.2 K per five years) and quite different, with some being positive and some being negative. It is not clear what these trends mean, except as an indication of the uncertainty of the radiosonde minus RO temperatures over this short time period. An estimate of the statistical significance of these trends would be useful. A comparison of actual (observed) and/or climate model temperature trends at these levels due to long-term climate change would be useful as well. For example, from climate models we might expect a temperature trend in the lower stratosphere to be something like 2-3 K per 100 years or $0.1 - 0.15$ K per five years. Trends reported in this paper for the Vaisala RS92 radiosonde at 50 hPa (Table 3) range from -0.211 K/5 years (U.S., night) to 0.264 K/5 years (England, day), so they are comparable or slightly larger than what one would expect for a long-term

climate trend signal.

It would also be interesting to compare these trends in radiosonde-RO temperature differences to the corresponding trends in the RO temperatures over this period. Indeed, Tables 3 and 4 give the RO trends, but they are never mentioned in the text. The RO temperature trends at 50 hPa (Table 3) range from -0.69 (Canada, day) to 1.143 (England, night). Quite different values are found at 150 hPa (Table 4), with the 5-year trends ranging from -0.797 (Canada, day) to 1.508 (U.S. day). In general, the magnitudes of the trends of radiosonde-RO temperature differences are smaller than the trends in RO temperatures, which is an indication of the consistency between the radiosonde and RO temperatures. The large differences in RO temperature trends between regions (much larger than expected for a long-term climate change signal) probably indicates natural variability in the six different regions. The fact that they are larger than the trends in the differences indicates to me that they represent a real signal in the different regions over this 7-year time period. Presumably, since the radiosonde-RO trends are smaller, the radiosondes (at least the most accurate ones) would pick up similar trends to the RO trends. A discussion of what the trends in radiosonde minus RO temperatures and RO temperatures means is needed.

In summary, the paper contains some interesting and important results and should be published, but it requires significant editing, and shortening with greater emphasis on what the important results are and less detail on all the individual numbers.
* * *

---

## Referee Comment (RC2) · Anonymous Referee #2 · 15 Dec 2016

This manuscript compares COSMIC and MetOP/GRAS GPS RO data with radiosonde temperatures in the interval 2006-2014. While this is not the first comparison of these data sets, the one presented here is an improvement due to the long time interval considered and because reprocessed COSMIC data have been used. The comparison is comprehensive and detailed. Accurate estimation of differences between radiosondes and GPS-RO is important since both are potentially used as "anchors" in reanalysis efforts. Other, less accurate data such as satellite radiances or aircraft temperatures are often bias-adjusted adaptively (Dee and Uppala 2009), whereas "anchors" are not.

Therefore I recommend eventual publication of the manuscript after undergoing the following major revisions: 1) Before the actual intercomparison, it should be specified what the structural uncertainties of the two measurement technologies are. These are mentioned for RO retrievals only in the supplement (+/-0.1 K in the 20-200 hPa range) For many modern radiosondes (RS92 in particular) they are specified as +/- 0.2 K below 100 hPa and somewhat higher at higher levels. RS-RO differences that fall within this range, especially if they are different in different regions of the world, should not be considered as "bias", as they may have other causes than systematic measurement errors. Small sample sizes or the different volumes sampled may be the reasons for the differences. 2) Modern radiosondes measure up to the 5hPa level, whereas this comparison stops at 20 hPa. Presumably this conservative choice is related to uncertainties in the inversion of the Abel integral necessary for the conversion of bending angles to refractivities. They lead to larger strucural uncertainties of the RO method. Could you elaborate on this, and also if the +/- 0.1K uncertainty specified for RO profiles applies to the 20 hPa level.

The other review points are minor: 3) The trend comparisons are difficult to interpret since the time interval is so short. Also the regional trend variability is much larger than the trend differences between RS and RO, at least for the more accurate radiosonde types.

4) When looking at the maps in Figure 2, it seems there is quite some heterogeneity even in countries with the same sensor, particularly at daytime, e.g. over China and Brasil. Can you give an explanation? It appears that the radiosonde type is not the only factor that determines the temperature biases. Do you think it is possible to estimate the biases also for each station individually? This has been done by several authors when homogenizing radiosonde time series. 5) The thresholds for daytime/nighttime (SZA < or > 90 deg) may not be optimal. Fig. 8c clearly shows positive biases at 90 deg, only at >95deg they are negative. Also the VIZ B2 and Shanghai sondes seem to reach their nighttime value at SZA clearly larger than 90 deg. I am also asking for

which times the SZA have been calculated? Nominal launch time of the radiosonde or time of collocation? Please clarify. 6) What is the reason for the strong decrease in measurement numbers already at 70 hPa over the US (Fig. 3a)? Did the reports from higher levels go missing? At most other places RS92 sondes consistently reach 20 hPa. 7) Fig. 3 onwards: You plot means and standard deviations. Instead you could plot means and the standard deviations of the MEAN (sigma^2/sqrt(N)) or 95% confidence intervals. This would allow a smaller scale for the x-axes. 8) Figs 5,8: Please triple number scale so that there is less intersection between number line and departures. 9) Fig. 9: Are these differences significant? The samples are smaller here. If std is 1.5K and number is 1000 for both samples, then the std of the means is roughly +/-0.05. For a 95% confidence interval you have to multiply with 1.96. Thus a large fraction of the differences shown in Fig. 9 would be insignificant. 10) Figs 11-13: Are the trends or trend differences significant? Please give confidence intervals for slopes. 11) Layer mean 20-200 hPa bias values in tables 1,2 are of limited use, since the biases changes a lot over this range of pressure. 12) Tables 3,4: What do you think is the reason for the very different biases over Brasil at 150 hPa for RS92 sondes? This level is well below the tropopause. Is it possible that water vapor or cloud content could adversely affect the RO estimates there. These effects have been neglected in Formula 1. 13) l718: traceability, not tractability. 14) l1385: RAOB instead of ROAB 15) l653-655: Some words are missing, the sentence does not seem to be complete. 16) l558: non-trivial

---

## Author Comment (AC1) · 24 Jan 2017

To reply reviewer's comments more specifically, also list the overall comments and specific comments and provide my responses below each individual comments. A pdf file for my responses is also submitted as a supplement.

Overall comments

The authors have done a lot of careful work and there are interesting results in their paper on the different accuracies, stabilities and trends of various types of radiosonde data. The authors use radio occultation (RO) data in the upper troposphere and lower

stratosphere as a reference data set for the period June 2006 to April 2014. Since radiosonde data are often used in climate studies, it is important to document the accuracies and uncertainties of different types of radiosondes in different countries.

1. However, the paper is too long (58 pages) and the sections on results (Sections 3-5) are tedious and difficult to read because of too much detail in the text that merely duplicates what is in the tables and figures as well as too many symbols in the text (e.g. △T (RS92200701-201012)). Furthermore, much of the detail is describing statistics that are small and probably not statistically significant or of general interest. The reader is overwhelmed with the reporting of many numbers without a focus on what is really important and what is of little or no interest. The paper would be greatly improved and have more impact if it were shortened significantly and only the major results included in the text.

âĞÍ As suggested by the reviewer, we shorten this paper significantly. We rewrote Section 3-5, combining section 4.2, section 4.3, and section 5 into a new section (new section 4.2). Many symbols (e.g. △T (RS92200701-201012)) are removed in the text. In addition, we removed results for 150 hPa (i.e., Tables 4 and 6) because the results were similar to those at 50 hPa. Appendix A is also removed, yet a part of the Appendix A is inserted in the Introduction section. To demonstrate whether the results are statistically significant or not, we performed statistical significance tests for RAOB – RO trend difference. We only mention that the statistical significance results in the revised paper. The revised manuscript is now reduced from 58 pages to 47 pages. All changes are tracked and the tracked manuscript is also submitted. In addition, since the heights from 150 hPa to 50 hPa are in part of upper troposphere, we added " "Upper Troposphere and" to the title.

2. There are many statistics of radiosonde minus RO temperatures from various types of radiosondes at different levels of the atmosphere between 200 and 20 hPa over six different regions of the world. It is not clear which of these results are statistically significant and which ones we should be concerned about. This makes interpretation of the

results difficult as we could be looking at small, quasi-random differences that have no physical meaning, nor even meaning relative to the specific types of radiosonde data. Differences are often 0.1K or less, which are well below the accuracy of radiosonde sensors. When the different atmospheric sampling volumes of the radiosondes and RO are considered, sampling errors alone can be much larger than 0.1K. It would be helpful if the authors could do statistical significance tests and describe in the text only the results that are significant at the 95% or higher level.

âĞÍ To demonstrate if the computed de-seasonalized trend differences (RAOB – RO) are statistically significant or not, we performed statistical significance tests for the trend difference. In Figs. 12 and 13 we list the 95% confidence intervals for trend difference (ROAB – RO) in the parentheses.

âĞÍ The 95% confidence intervals for trend differences for global Vaisala (RS80, RS90, and RS92) and other sensor types during the daytime and night time in the North hemisphere mid-latitude (60°N-20°N) are summarized in the Table 4. We also discuss what the trends in radiosonde minus RO temperatures and RO temperatures means is the text (see the reply for comment 5).

3. The authors compute trends of the differences between individual types of radiosonde and RO over a 7-year period. Most of the trends are small (of order 0.2 K per five years) and quite different, with some being positive and some being negative. It is not clear what these trends mean, except as an indication of the uncertainty of the radiosonde minus RO temperatures over this short period. Again an estimate of the statistical significance of these trends would be useful. What would the magnitude of trends computed from a similar time series of random data with the same standard deviation as these differences be? A comparison of what sort of trends in temperature at these levels due to long-term climate change would be useful as well. For example, from climate models we might expect a temperature trend in the lower stratosphere to be something like 5 K per 100 years or 0.25 K per five years. Trends reported in this paper for the Vaisala RS92 radiosonde at 50 hPa (Table 3) range from -0.211 K/5

years (U.S., night) to 0.264 K/5 years (England, day), so they are comparable or slightly smaller than what one would expect for a long-term climate trend signal.

⇨ To compare trends in temperature at these levels due to long-term climate change with RO trends in this paper, we refer to stratospheric temperature trends over 1979–2015 computed by Randel et al., (2016) in line 466. Randel et al., (2016) is also added in the references. Randel et al., (2016) indicated that the linear trends over 1979–2015 show that cooling in the lower stratosphere is about -0.1 K to 0.2 K/decade. In line 464 we added, "A long-term (de-seasonalized) trend in temperature at this level associated with global warming (stratospheric cooling) might be approximately -0.1 to -0.2 K/decade or -0.05 to -0.1 K/5 years (Randel et al., 2016). Trends reported in this paper for the Vaisala RS92 radiosonde at 50 hPa (Table 3) range from -0.211 K/5 years (U.S., night) to 0.264 K/5 years (United Kingdom, day), which are comparable to those reported by Randel et al., (2016)." 4. It would also be interesting to compare these trends in radiosonde-RO temperature differences to the corresponding trends in the RO temperatures over this period. Indeed, Tables 3 and 4 give the RO trends, but they are never mentioned in the text!

⇨ As suggested by the reviewer, we rewrote sections 4 and 5 and compared de-seasonalized trends of radiosonde-RO temperature differences to the corresponding trends in RO temperatures over this period.

⇨ In line 455 we added "The de-seasonalized trends in RO temperatures are generally larger than those for the radiosonde-RO differences. A maximum de-seasonalized trend of 1.143 K/5 yrs is found for nighttime temperatures over the United Kingdom. A minimum de-seasonalized trend of -0.69 K/5 yrs is found for daytime temperatures over Canada. Trends with magnitude greater than 0.5 K/5 yrs are found over the United States, Germany, Canada and the United Kingdom. The fact that these de-seasonalized trends in RO are significantly greater than the de-seasonalized trends in the differences suggests that they represent a physical signal in these regions. However, the time series is too short to represent a long-term climate signal; instead these

likely represent real but short-term trends associated with natural variability."

âĞÍ To shorten the paper, we also removed results for RAOB and RO temperature comparisons at 150 hPa (old Tables 4 and 6). Now the new Tables 3 and 4 are specifically mentioned in the new text.

5. The RO temperature trends at 50 hPa (Table 3) range from -0.69 (Canada, day) to 1.143 (England, night). Quite different values are found at 150 hPa (Table 4), with the 5-year trends ranging from -0.797 (Canada, day) to 1.508 (U.S. day). In general, the magnitudes of the trends of radiosonde-RO temperature differences are smaller than the trends in RO temperatures, which is an indication of the consistency between the radiosonde and RO temperatures. The large differences in RO temperature trends between regions (much larger than expected for a long-term climate change signal) probably indicates natural variability in the six different regions. The fact that they are larger than the trends in the differences indicates to me that they are a real signal in the different regions over this 7-year time period. Presumably, since the radiosonde-RO trends are smaller, the radiosondes (at least the good ones) would pick up similar trends to the RO trends. A discussion of what the trends in radiosonde minus RO temperatures and RO temperatures means is needed.

âĞÍ We specifically added several discussions of what the trends in radiosonde minus RO temperatures and RO temperatures means in the new Section 4.2.

As mentioned in the reply for comment 4, we added a discussion of what the trends in radiosonde minus RO temperatures and RO temperatures means in Line 455. In line 460, we specifically discuss what the trends in radiosonde minus RO temperatures and RO temperatures means: "The fact that these de-seasonalized trends in RO are significantly greater than the de-seasonalized trends in the differences suggests that they represent a physical signal in these regions. However, the time series is too short to represent a long-term climate signal; instead these likely represent real but short-term trends associated with natural variability. "

âĞÍ In lines 470-480, we discuss what trends in radiosonde minus RO temperatures and RO temperatures means by stating "We compare the global trend of radiosonde – RO temperature differences for the Vaisala and other radiosondes at 50 hPa in Table 4. The Vaisala RS92 biases are 0.22 K (day) and 0.12 K (night). The global de-seasonalized temperature differences for Vaisala RS92 for daytime and nighttime are equal to 0.074 K/5yrs and -0.094 K/5yrs, respectively. The 95% confidence intervals for slopes are shown in the parentheses in Table 4. This indicates that although there might be a small residual radiation error for RS92, the trend in RS92 and RO temperature differences from June 2006 to April 2014 is within +/-0.09 K/5yrs globally. These values are just above the 1-sigma calibration uncertainty estimated by Dirksen et al. (2014). This means that probably the stability of the calibration alone could explain most of this very small trend. It is also consistent with the change in radiation correction."

âĞÍ We discuss the mean bias in the last two paragraphs of Section 4.2. In line 481 we state "Figure 13 depicts the de-seasonalized temperature differences for Sippican MARK IIA, VIZ-B2, AVK-MRZ, and Shanghai in North hemisphere mid-latitude (60°N-20°N) at 50 hPa and the results are summarized in Table 4. The 95% confidence intervals for slopes are shown in the parentheses in Table 4. The de-seasonalized trend of the daytime differences varies from -0.137 K/5 years (Russia) to 0.468 K/5 years (VIZ-B2). The magnitudes of the daytime trends are less than 0.2 K/5 yrs for all sensor types except VIZ-B2 and Sippican, which both exceed 0.4 K/5 yrs. These are much larger than those of the Vaisala RS92 (0.074 K/5 yrs)."

âĞÍ In line 489, we state, "The corresponding nighttime de-seasonalized trends in the biases vary from -0.348 K/5 yrs (VIZ-B2) to 0.244 K/5 yrs (Sippican). Again, these are much larger than those of Vaisala RS92 (-0.094 K/5 yrs). Thus the VIZ-B2 sensor stands out as having larger biases and trends than do the other sensors."

In summary, the paper contains some interesting and important results and should be published, but it requires significant rewriting, editing, and shortening with greater emphasis on what the important results are and less detail on all the individual numbers.

Detailed comments

1. The papers use three terms to describe the radiosonde-RO temperature differences: differences, biases, and anomalies. I suggest using only differences and biases, and eliminate all references to anomalies.

âĞÍ We replace all "anomalies" with "differences" in this paper.

2. Do you mean United Kingdom rather than England?

âĞÍ Yes, it shall be "United Kingdom". We replace all "England" with "United Kingdom" in this paper and Figures. The revised Figs. 3, 4, 5, 9, 10, and 12 are inserted.

3. An example of how a difficult to read paragraph containing a repetition of data in a table can be simplified, shortened, and made more readable is lines 264-267:

"In general, the radiosonde temperature biases vary for different sensor types. The mean $\Delta T$ for RS92 (0.16 K), RS80 (0.10 K), RS90 (0.13 K), Sippican MarkIIA (-0.08 K), Shangai (0.05 K) and Meisei (0.11 K) are smaller than those for AVK (0.33 K) and VIZ-B2 (0.22 K) (see Table 2)" (50 words) may be replaced with

"The radiosonde temperature biases vary for different sensor types. All biases are less than 0.25 K, except for AVK and VIZ-B2, which reach 0.66 and 0.71 K respectively during the day." (31 words).

âĞÍ The sentence in lines 264-267 was revised as suggested by the reviewer. Similar sentences in old Sections 3-5 were also revised and are not specifically mentioned. The tracked manuscript is submitted.

4. An example of unnecessary use of symbols in a sentence which makes reading difficult is: "The mean temperature biases in this region for , , , and for 50 hPa are summarized in Table 5." This can be written in much more readable form as "The mean temperature biases in this region for the Sippican, VIZ-B2, AVK, and Shanghai

radiosondes at 50 hPa are summarized in Table 5."

âĞÍ We removed many symbols (for example, ) in this paper and many sentences are revised as suggested. For example, in line 426, we now state "All daytime biases are below 0.25 K in magnitude, except for Russia (0.8 K) and VIS-B2 (0.87 K). The magnitudes of the mean nighttime biases are all less 0.25 K except for VIS-B2, which is -0.56 K. The daytime biases for Russia and VIS-B2 contain obvious inter-seasonal variation." No symbols are used in the sentence.

5. Lines 439-442. I don't understand this sentence. If the U.S. did not use RS92 radiosondes before 2012, there would be no data for comparison with RO before 2012 (i.e. none in the period 2007-2010). However, this section talks about RS92 from Jan 2007 to Dec 2010 for the U.S. and Fig. 10 shows RS92 vs. RO going back to 2007. Also, a small number of pairs in the comparison does not necessarily imply small differences—in fact, a small number of pairs could lead to large differences due to an inadequate sample size.

âĞÍ The US National Weather Service (NWS) did not use Vaisala RS92 radiosondes before 2012. The Vaisala RS92 radiosondes before 2012 were mainly launched by research groups (for example, at the ARM site and during individual field experiments from universities, etc).

âĞÍ To avoid confusion by the readers, we deleted the statement "since the US National Weather Service (NWS) did not use Vaisala RS92 radiosondes before 2012."

6. It seems strange that Table 3 is not mentioned until line 563, long after Tables 4, 5 and 6 are mentioned and discussed.

âĞÍ In the revised paper, we refer to Table 3 before Table 4. The Table 3 is now referred in line 415 whereas Table 4 is first referred in line 426.

Please also note the supplement to this comment:

[Figure]

http://www.atmos-chem-phys-discuss.net/acp-2016-801/acp-2016-801-AC1-supplement.pdf

---

## Author Comment (AC2) · 24 Jan 2017

This manuscript compares COSMIC and MetOP/GRAS GPS RO data with radiosonde temperatures in the interval 2006-2014. While this is not the first comparison of these data sets, the one presented here is an improvement due to the long time interval considered and because reprocessed COSMIC data have been used. The comparison is comprehensive and detailed. Accurate estimation of differences between radiosondes and GPS-RO is important since both are potentially used as "anchors" in reanalysis efforts. Other, less accurate data such as satellite radiances or aircraft temperatures

are often bias-adjusted adaptively (Dee and Uppala 2009), whereas "anchors" are not.

Therefore I recommend eventual publication of the manuscript after undergoing the following major revisions:

1) Before the actual intercomparison, it should be specified what the structural uncertainties of the two measurement technologies are. These are mentioned for RO retrievals only in the supplement (+/-0.1 K in the 20-200 hPa range). For many modern radiosondes (RS92 in particular) they are specified as +/- 0.2 K below 100 hPa and somewhat higher at higher levels. RS-RO differences that fall within this range, especially if they are different in different regions of the world, should not be considered as "bias", as they may have other causes than systematic measurement errors. Small sample sizes or the different volumes sampled may be the reasons for the differences.

=> As suggested by the reviewer#1, we shortened this paper significantly. In the revised paper line 261, we added "For many modern radiosondes (for example RS92) the structural uncertainties are +/- 0.2 K below 100 hPa and somewhat higher at higher levels."

2) Modern radiosondes measure up to the 5hPa level, whereas this comparison stops at 20 hPa. Presumably this conservative choice is related to uncertainties in the inversion of the Abel integral necessary for the conversion of bending angles to refractivities. They lead to larger strucural uncertainties of the RO method. Could you elaborate on this, and also if the +/- 0.1K uncertainty specified for RO profiles applies to the 20 hPa level.

=> In the revised paper, we moved parts of Appendix A (The Quality of GPS RO Data as Benchmark References and the New Reprocessed Package) to the introduction section. In line 112, we specifically stated "At 20 hPa, the mean temperature difference between COSMIC and CHAMP was within 0.05K (Ho et al., 2009b)." => In line 113 we specially stated the mean layer temperature difference between 200 hPa to 10 hPa is within 0.05 K, and at 20 hPa, the mean temperature difference is equal to 0.03

K:" Schreiner et al. (2014) compared re-processed COSMIC and Metop-A/GRAS (Meteorological Operational Polar Satellite–A/Global Navigation Satellite System (GNSS) receiver for Atmospheric Sounding) bending angles and temperatures produced at COSMIC Data Analysis and Archive Center (CDAAC). The mean layer temperature difference between 200 hPa to 10 hPa was within 0.05 K where the mean temperature difference at 20 hPa is equal to 0.03K. This demonstrates the consistency of COSMIC and Metop-A/GRAS temperatures."

=> The current results are for 200 hPa and 20 hPa, where the ionospheric effect is minimal.

=> To estimate the uncertainty of RO temperatures in the upper troposphere and lower stratosphere, particularly between 200 hPa and 10 hPa, we stated in line 122: "To estimate the uncertainty of RO temperature in the upper troposphere and lower stratosphere, Ho et al., (2010) compared RO temperature from 200 hPa to 10 hPa to those from Vaisala-RS92 in 2007 where more than 10,000 pairs of coincident Vaisala-RS92 and COSMIC data were collected. The mean bias in this height range was equal to -0.01 K with a mean standard deviation of 2.09 K. At 20 hPa, the mean bias was equal to -0.02K. These comparisons demonstrate the quality of RO temperature profiles in this height range."

=> Based on above studies, we are confident that an uncertainty of +/- 0.1K for RO profiles does apply to the 20 hPa level.

The other review points are minor:

3) The trend comparisons are difficult to interpret since the time interval is so short. Also the regional trend variability is much larger than the trend differences between RS and RO, at least for the more accurate radiosonde types.

=> As suggested by the reviewer#1, we shortened this paper significantly. We rewrote sections 3-5, combining section 4.2, section 4.3, and section 5 into a new section (new

section 4.2). In the new section 4.2, we add several paragraphs to discuss what the trends in radiosonde minus RO temperatures and RO temperatures means.

As mentioned in the reply for comment 4, we added a discussion of what the trends in radiosonde minus RO temperatures and RO temperatures means in Line 455. In line 460, we discuss what the trends in radiosonde minus RO temperatures and RO temperatures mean: "The fact that these de-seasonalized trends in RO are significantly greater than the de-seasonalized trends in the differences suggests that they represent a physical signal in these regions. However, the time series is too short to represent a long-term climate signal; instead these likely represent real but short-term trends associated with natural variability."

=> In lines 470, we discuss what the trends in radiosonde minus RO temperatures and RO temperatures mean by stating "We compare the global trend of radiosonde – RO temperature differences for the Vaisala and other radiosondes at 50 hPa in Table 4. The Vaisala RS92 biases are 0.22 K (day) and 0.12 K (night). The global de-seasonalized temperature differences for Vaisala RS92 for daytime and nighttime are equal to 0.074 K/5yrs and -0.094 K/5yrs, respectively. The 95% confidence intervals for slopes are shown in the parentheses in Table 4. This indicates that although there might be a small residual radiation error for RS92, the trend in RS92 and RO temperature differences from June 2006 to April 2014 is within +/-0.09 K/5yrs globally. These values are just above the 1-sigma calibration uncertainty estimated by Dirksen et al. (2014). This means that probably the stability of the calibration alone could explain most of this very small trend. It is also consistent with the change in radiation correction."

=> We discuss the mean bias in the last two paragraphs of Section 4.2. In line 481 we stated "Figure 13 depicts the de-seasonalized temperature differences for Sippican MARK IIA, VIZ-B2, AVK-MRZ, and Shanghai in North hemisphere mid-latitude (60°N-20°N) at 50 hPa and the results are summarized in Table 4. The 95% confidence intervals for slopes are shown in the parentheses in Table 4. The de-seasonalized trend of the daytime differences varies from -0.137 K/5 years (Russia) to 0.468 K/5

years (VIZ-B2). The magnitudes of the daytime trends are less than 0.2 K/5 yrs for all sensor types except for VIZ-B2 and Sippican, both of which exceed 0.4 K/5 yrs. These are much larger than those of the Vaisala RS92 (0.074 K/5 yrs)."

=> In line 489, we stated "The corresponding nighttime de-seasonalized trends in the biases vary from -0.348 K/5 yrs (VIZ-B2) to 0.244 K/5 yrs (Sippican). Again, these are much larger than those of Vaisala RS92 (-0.094 K/5 yrs). Thus the VIZ-B2 sensor stands out as having larger biases and trends than do the other sensors."

4) When looking at the maps in Figure 2, it seems there is quite some heterogeneity even in countries with the same sensor, particularly at daytime, e.g. over China and Brasil. Can you give an explanation? It appears that the radiosonde type is not the only factor that determines the temperature biases. Do you think it is possible to estimate the biases also for each station individually? This has been done by several authors when homogenizing radiosonde time series.

=> We suspect that the heterogeneity over China may be due to inconsistent corrections applied in northern and southern provinces of China. In general, the Chinese sondes contain their corrections, which are not documented in public literature.

=> The heterogeneity over Brazil may be due to a smaller sample size at certain stations. For example, we found that stations with temperature biases larger than 0.5 K in the east Brazil contain only about 60 RO-RAOB pairs.

=> Two more sentences were added in section 3.1 "Although we only include stations containing more than 50 RO-RAOB pairs, some level of heterogeneity (i.e., Fig. 2a over Brazil) may be due to low sample numbers. For example, stations with temperature biases larger than 0.5 K in eastern Brazil contain only about 60 RO-RAOB pairs. The cause of the heterogeneity in temperature bias between North and South China is not certain at this point."

5) The thresholds for daytime/nighttime (SZA < or > 90 deg) may not be optimal. Fig.

8c clearly shows positive biases at 90 deg, only at >95deg they are negative. Also the VIZ B2 and Shanghai sondes seem to reach their nighttime value at SZA clearly larger than 90 deg. I am also asking for which times the SZA have been calculated? Nominal launch time of the radiosonde or time of collocation? Please clarify.

=> We tested several criteria and decided to use thresholds of SZA < 90 as daytime and SZA > 90 as nighttime. It is possible that this could contain scattering effects during dawn and sunset, but we given the uncertainties of the actual time of observations, this threshold appeared most appropriate.

=> The SZA is computed from the launch time and location of sonde station since the information of specific time and location of sonde at different height is not available.

=> We added: "The SZA is computed from the synoptic launch time and location of sonde station since the time and location of the sonde at different height is not available." in the end of section 2.3.

6) What is the reason for the strong decrease in measurement numbers already at 70 hPa over the US (Fig. 3a)? Did the reports from higher levels go missing? At most other places RS92 sondes consistently reach 20 hPa.

=> The height is determined by the balloon used at various sites. There are about 15 stations launching RS92 during the study period. Our best guess is that these US stations are only interested in the tropospheric profiles and use smaller balloons. Meteorological services usually try to get to 50 or 30 hPa for all soundings and use slightly larger balloons. GRUAN stations are required to reach 5 hPa and should use larger balloons. The sondes launched ARM site also reach to 5 hPa.

=> To provide the possible reason for the strong decrease in measurement numbers already at 70 hPa over the US (Fig. 3a), in Line 314 we added "Figure 3 indicates that RS92 in different regions demonstrate a similar quality in terms of mean differences from RO with a small warm bias above 100 hPa, as well as similar standard deviations

relative to the mean biases of approximately 1.5K. Because some stations in the United States are only interested in the tropospheric profiles and use smaller balloons, fewer RO-RS92 samples are available above 70 hPa compared to those in other countries."

7) Fig. 3 onwards: You plot means and standard deviations. Instead you could plot means and the standard deviations of the MEAN (sigmaËĘ2/sqrt(N)) or 95% confidence intervals. This would allow a smaller scale for the x-axes.

=> I think the review is talking about "standard error of the mean". We did plot the standard error of the mean. Since there are a lots of RO-RAOB pairs, the standard error of the mean is too small to see.

=> The standard deviation of the mean are plotted in current Figs. 3, 4, 6, 7. We also add the standard error of the mean in Figs. 3, 4, 6, and 7. To make it clear, we re-state "We also plot the standard error of the mean (black dot) superimposed on the mean. The value of the standard error of the mean is less than 0.03 K depending on the sample numbers " in the caption of Fig. 3.

8) Figs 5,8: Please triple number scale so that there is less intersection between number line and departures.

=> The number scale in Figs. 5 and 8 is revised. The new Figs. 5 and 8 are used in the paper.

9) Fig. 9: Are these differences significant? The samples are smaller here. If std is 1.5K and number is 1000 for both samples, then the std of the means is roughly +/-0.05. For a 95% confidence interval you have to multiply with 1.96. Thus a large fraction of the differences shown in Fig. 9 would be insignificant.

=> The purpose of Fig. 9 is to use RO temperature as references to identify the RS92 temperature biases due to change of radiation correction. With the uncertainty of RO data (+/- 0.1K uncertainty) and RAOB data ((+/- 0.2K uncertainty below 100 hPa and larger uncertainty above that), it is hard to say the results are significant.

=> Therefore, we added a new paragraph in line 387 "There is no consistent pattern of differences in these two periods over the six regions, with mean differences ranging from -0.122 K (Australia) to 0.047 K (United States). The small differences in profile shapes and magnitudes are an indication of the magnitude of the uncertainty in RS92 temperatures due to differences in implementing the radiation correction tables."

10) Figs 11-13: Are the trends or trend differences significant? Please give confidence intervals for slopes.

=> Confidence intervals for slopes are added in each panel in Figs. 12 and 13. The confidence intervals for slopes are shown in the parentheses in each panels of Figs. 12 and 13.

11) Layer mean 20-200 hPa bias values in tables 1,2 are of limited use, since the biases changes a lot over this range of pressure.

=> Table 2 summarizes the change of the mean and standard deviation of temperature differences (K) between 200 hPa and 20 hPa between RO and eight types of radiosonde. This is to demonstrate that RO temperature can be used as references to distinguish the temperature biases among sonde types and their biases at daytime and nighttime of the comparison in the rest of the paper. We think this is important and we will keep Tables 1 and 2.

12) Tables 3,4: What do you think is the reason for the very different biases over Brasil at 150 hPa for RS92 sondes? This level is well below the tropopause. Is it possible that water vapor or cloud content could adversely affect the RO estimates there. These effects have been neglected in Formula 1.

=> In the revised paper we remove results for those for 150 hPa (i.e., old Tables 4 and 6) because the results were similar to those at 50 hPa.

=> The reason for the larger biases over Brazil for RS92 may be due to the incomplete bias correction. Figure 5 shows that the mean $\Delta T$ (RS92) has a slightly larger warm

bias for low SZA (near noon) than that at higher SZA (late afternoon and in the night). The mean SZA for the RO-RS92 pairs over USA, Canada, and Brazil are 64.7 degree, 78.4 degree, and 45.9 degree, respectively. Because daytime SZA over Brazil is in general smaller (close to the noon) than other regions, the Brazil temperature biases relative to the collocated RO data are higher than other regions.

=> The attached figure (this will not be shown in the paper) depicts the seasonal variation of SZA over different regions.

This figure is inserted in the supplement. This figure is also submitted as Figure 1.

13) l718: traceability, not tractability.

=> In line 718 and 491, the "tractability" is replaced by "traceability "

14) l1385: RAOB instead of ROAB

=> In line 11385, the "ROAB" is replaced by "RAOB"

15) l653-655: Some words are missing, the sentence does not seem to be complete.

=> The sentence is completed now.

16) l558: non-trivial

=> In line 1558, "non-trivial" is added.

Please also note the supplement to this comment:
http://www.atmos-chem-phys-discuss.net/acp-2016-801/acp-2016-801-AC2-supplement.pdf

———————————————————

[Figure]

**Fig. 1.**

---

## Author Response (AR2)

**Responses to editor and reviewers' comments**

**Editor's Comments to the Author:**

Based on the reviews of your revised version I am happy to accept your paper for publication in ACP. Congratulations. Please take the technical comments into account when preparing the final version.

Rolf Müller

Non-public comments to the Author:
I have noticed:

l). 505: Russian --> Russia

⇨ "Russian" is replaced by "Russia"

2). 506: Stated --> States

⇨ "Stated" is replaced by "States"

**Comments from Reviewer#1**

1). The paper is much improved over the previous version. It is shorter and clearer and the conclusions are for the most part well stated. The main issue that needs to be addressed in the final version is to be consistent in the order of radio occultation (RO) and radiosonde when talking about differences and biases. In all cases of the reported numerical differences, it is "radiosonde minus RO," but in some places in the text (including the abstract), "RO minus radiosonde" is used. This is confusing. "Radiosonde minus RO" and "radiosonde-RO" should be used throughout the paper.

⇨ All the order of radio occultation (RO) and radiosonde when talking about differences and biases are rewritten as "radiosonde minus RO" throughout the paper.

2). There are many numbers reported in the paper, and sometimes it is not clear which figure or table they come from. This should be clarified in the final text.

⇨ All the numbers reported in this paper are double checked. For example, to make sure the numbers in the text agree with those listed in Tables and Figures, we re-write line 44 (in abstract) as "The mean radiosonde-RO global daytime temperature difference in the layer from 200 hPa to 20 hPa for Vaisala RS92 is equal to 0.20 K. The corresponding difference is equal to -0.06 K for Sippican, 0.71 K for VIZ-B2, 0.66 K for Russian AVK-MRZ, and 0.18 K for Shanghai."

⇨ We also insert several referred Tables and Figures when those numbers are mentioned in the text so that readers will know where exactly those numbers are coming from.

3). Finally, there are a number of minor editorial issues and typographical error that I will not summarize here. Instead, I sent suggest edits directly to the authors.

⇨ We had gone through another round of editing. Several typographical errors are corrected. The tracked and revised version is also submitted.

**Comments from Reviewer#2**

The paper has considerably gained in quality. There are only a few minor points:

1) Please do not give 5yr trends in 0.001K/5yr precision when the confidence interval are more than 0.1K/fyr. 0.01/5yr is already too much but could be tolerated

⇨ We agree to report precision level of 0.01K/5 yrs in the text but in this stage we do not further revise the numbers in Tables and Figures.

⇨ The trend "5 years" are replaced by "5 yrs" throughout the paper.

2) Abstract line 41: Please specify layer to which the mean applies (20-200 hPa I presume). This is important.

⇨ In line 44 we add " in the layer from 200 hPa to 20 hPa for Vaisala RS92"

3) l95: research, not "researches"

⇨ In line 70 "researches" is replaced by "research"

4) l430: this is not an adequate citation. Perhaps the link is gone the time the paper is published.

=> I assume the review meant link in line 422 in the revised paper "http://www.vaisala.com/en/products/soundingsystemsandradiosondes/soundingdatacontinuity/RS92DataContinuity/Pages/revisedsolarradiationcorrectiontableRSN2010.aspx".

This is still a correct link.

5) l474: low sample sizes, not sample "numbers"

=> In line 317, we replace "sample numbers" with "sample sizes"

6) l530: Figure 3 indicates that RS92 measurements have high quality in terms...

⇨ In line 348, we revise that sentence as "Figure 3 indicates that RS92 measurements in different regions have a similar quality in terms of"

7) l856: less than 0.25

⇨ In line 471, we add "less than 0.25K"